# When Does Adaptation Win? Scaling Laws for Meta-Learning in Quantum Control

**Nima Leclerc**[1]   **Chris Miller**[1]   **Nicholas Brawand**[1]

## Abstract

Quantum hardware suffers from intrinsic device heterogeneity and environmental drift, forcing practitioners to choose between suboptimal non-adaptive controllers or costly per-device recalibration. We derive a scaling law lower bound for meta-learning showing that the adaptation gain (expected fidelity improvement from task-specific gradient steps) saturates exponentially with gradient steps and scales linearly with task variance, providing a quantitative criterion for when adaptation justifies its overhead. Validation on quantum gate calibration shows negligible benefits for low-variance tasks but $> 40\%$ fidelity gains on two-qubit gates under extreme out-of-distribution conditions ($10\times$ the training noise), with implications for reducing per-device calibration time on cloud quantum processors. Further validation on classical linear-quadratic control confirms these laws emerge from general optimization geometry rather than quantum-specific physics. We further introduce a few-shot pre-adaptation protocol that estimates the optimal adaptation budget from $N=3$–$5$ probe steps within 3–19% relative error across out-of-distribution regimes. ⌾ Code.

## 1. Introduction

Quantum processors face a persistent calibration bottleneck: current devices could easily require 2 hours of calibration daily for a 100-qubit system (Mohseni et al., 2024), with qubit relaxation time ($T_1$) variations of 20–60 $\mu$s (Burnett et al., 2019) and time-dependent noise sources (Berritta et al., 2025), necessitating frequent recalibration to maintain a desired gate *fidelity* (how closely a gate matches its ideal implementation) (Krinner et al., 2020). Calibration involves

optimizing the microwave or laser control pulses that implement quantum gates, a process that must be repeated as system parameters drift.

As quantum computers scale to thousands of qubits, this overhead becomes prohibitive. For example, IBM's quantum systems require daily calibrations lasting 30–90 minutes depending on system size, with additional hourly recalibrations (IBM Quantum, 2025).

Practitioners face two broad strategies for managing device heterogeneity. Standard approaches like Gradient Ascent Pulse Engineering (GRAPE) (Khaneja et al., 2005) optimize a single robust or non-adaptive pulse sequence for assumed noise conditions but require accurate noise models and lengthy re-optimization when those assumptions fail. Adaptive approaches have emerged more recently: reinforcement learning (RL) methods learn device-specific control policies (Bukov et al., 2018; Niu et al., 2019; Ernst et al., 2025) but can be sample-inefficient, while meta-learning frameworks like metaQctrl (Zhang et al., 2025) demonstrate that learned initializations can improve robustness under parameter uncertainty. However, existing work does not characterize *when* adaptation outperforms non-adaptive deployment or *how* adaptation gains scale with task variance. We provide this characterization. Throughout this paper, we refer to a *task* $\xi$ as a distinct calibration instance, characterized by device-specific noise rates, coherence times, or coupling strengths that vary due to fabrication differences or environmental drift. Figure 1 previews our framing: heterogeneity across devices (a) is absorbed into a meta-initialization trained on a differentiable simulator (b), then specialized per-device at deployment (c).

Similar tradeoffs arise in robotics and fusion control (Seo et al., 2024; Chen et al., 2025), but quantum systems offer unique advantages as a theoretical testbed: the dynamics are exactly specified by the Lindblad master equation (Gorini et al., 1976; Lindblad, 1976), and controllability is analytically characterized (O'Meara, 2025), providing ground-truth dynamics for evaluating any control policy under arbitrary environmental conditions. We assume that noise statistics are stationary; non-stationary control drift within an episode would require online adaptation extensions.

---

[1]Quantum Information Sciences, Optics, and Imaging Department, The MITRE Corporation, 7525 Colshire Dr, McLean, VA, USA 22102. Correspondence to: Nima Leclerc <nleclerc@mitre.org>.

*Proceedings of the $43^{rd}$ International Conference on Machine Learning*, Seoul, South Korea. PMLR 306, 2026. Copyright 2026 by the author(s).

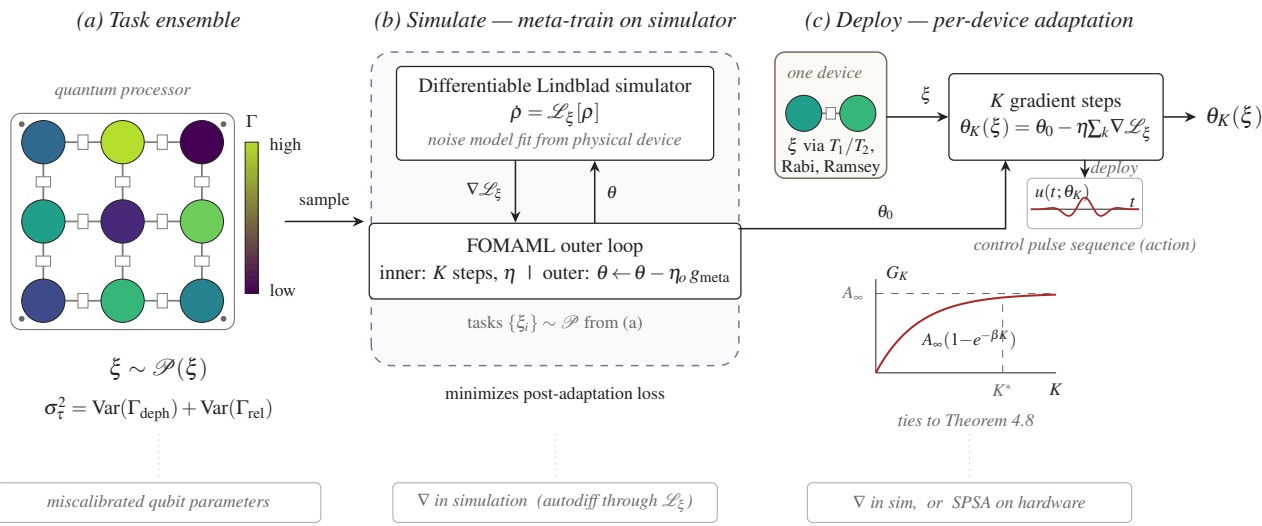

Figure 1. **Simulate-then-deploy workflow for meta-learned quantum control.** (a) A heterogeneous processor of transmons with noise rates $\xi = (\Gamma_{\text{deph}}, \Gamma_{\text{relax}}) \sim \mathcal{P}(\xi)$; heterogeneity is summarized by $\sigma_\tau^2 = \text{Var}(\Gamma_{\text{deph}}) + \text{Var}(\Gamma_{\text{rel}})$. (b) A differentiable Lindblad simulator ($\dot{\rho} = \mathcal{L}_\xi[\rho]$), with noise model fit from device characterization, is meta-trained via first-order model-agnostic meta-learning (FOMAML) on tasks $\{\xi_i\} \sim \mathcal{P}$ to yield initialization $\theta_0$. $\eta_0$ and $g_{\text{meta}}$ are the outer loop learning rate and meta-gradient. (c) At deployment, relaxation and dephasing times ($T_1$ and $T_2$) via Rabi and Ramsey characterization supplies $\xi$; $K$ gradient steps produce $\theta_K(\xi)$ parameterizing the emitted control pulse $u(t; \theta_K)$. The saturation curve $G_K = A_\infty(1 - e^{-\beta K})$ (inset) characterizes the adaptation gain, with $K^*$ marking diminishing returns.

Crucially, universal quantum computation decomposes into single-qubit rotations and two-qubit entangling gates (Nielsen & Chuang, 2010). Any quantum algorithm, regardless of circuit depth or qubit count, reduces to repeated application of these primitive operations. The calibration bottleneck therefore scales with processor size: a 100-qubit processor requires calibrating approximately 100 single-qubit gate sets and $O(n)$ two-qubit couplers (Arute et al., 2019), each facing device-specific drift and noise variation. Our scaling laws address precisely this operational challenge, providing quantitative criteria for when per-gate adaptation justifies its overhead. Figure 2(a) illustrates this tradeoff geometrically. In control parameter space, the non-adaptive controller (designed to be robust to the average noise condition) $\theta_{\text{rob}}^*$ (star) minimizes the expected loss across all tasks but lies away from the task-specific optima (red and green dots). Gradient-based adaptation traces distinct trajectories from this shared initialization by taking $K$ steps toward each device's optimum, yielding task-specific pulse sequences (colored red and green, corresponding to each task). The question is whether these $K$ gradient steps justify their computational cost.

We derive scaling laws for the *adaptation gap*, defined as the expected fidelity improvement when adapting a meta-learned initialization $\theta_0$ with $K$ steps to a specific device parameterized by $\xi$: $G_K = \mathbb{E}_\xi[L_\xi(\theta_0) - L_\xi(\theta_K)]$, where $L_\xi(\theta)$ denotes the policy's loss function for weights $\theta$ for task $\xi$. This gap satisfies $G_K \geq c\sigma_\tau^2(1 - e^{-\beta K})$,

linking task variance ($\sigma_\tau^2$), adaptation rate ($\beta$), and gradient steps ($K$) to gains over the robust baseline.

**Key assumption.** Our analysis requires the Polyak-Łojasiewicz (PL) condition (Karimi et al., 2016) locally near optima, guaranteeing that gradient steps yield predictable improvement in the loss. For closed quantum systems, trap-free landscapes (Russell et al., 2017) (no local minima other than the global optimum) suggest favorable optimization geometry; for open systems, we validate the PL condition empirically (Figure 3 (a)), with PL satisfaction nearly universal at moderate thresholds across in-distribution (ID) and strong out-of-distribution (OOD) regimes; even when the strict threshold is violated, the exponential fit on $G_K$ retains $R^2 > 0.95$ (Appendix A.7). This assumption implies that the meta-initialization is already near the basin of attraction. The resulting adaptation gap satisfies $G_K \geq A_\infty(1 - e^{-\beta K})$, with a polynomial-decay generalization under the weaker Kurdyka–Łojasiewicz (KL) condition (Appendix C.2). As illustrated in Figure 2(b), the gap exhibits exponential saturation at rate $\beta = \eta\mu$, where $\eta$ is the inner-loop learning rate (Finn et al., 2017) and $\mu$ characterizes the local landscape curvature, with asymptote $A_\infty \propto \sigma_\tau^2$ scaling with device-to-device variability. We validate our predictions on single-qubit gates and two-qubit entangling gates, demonstrating strong quantitative agreement ($R^2 \geqslant 0.98$), which provides empirically-validated guidelines for when adaptation provides meaningful benefit over a fixed controller in gate calibration.

**Contributions**

- **Scaling laws for quantum gate calibration.** We derive the bound $G_K \geq A_\infty(1 - e^{-\beta K})$ on the adaptation gap, with a polynomial-decay analogue under the KL condition. Here $\beta = \eta\mu$ captures adaptation rate via step size ($\eta$) and curvature ($\mu$); $A_\infty$ quantifies device variability.

- **Validation on universal gate primitives.** On differentiable quantum simulators, exponential saturation holds for both single-qubit $X$ gates and two-qubit CZ gates, with >40% fidelity improvement under high-noise OOD conditions. These form a universal gate set ([Barenco et al., 1995](#)).

- **Operational decision protocol.** A few-shot procedure estimates the 95%-gain budget $\hat{K}_{0.95}$ within 3–19% across 2×–5× OOD regimes. A companion ablation shows adaptation alone (without $\xi$) drives most of the recovery under shift (Section 5.4).

We position the framework as a *simulate-then-deploy* paradigm (Figure 1): scaling laws are characterized in simulation, and the meta-initialization is deployed to hardware where on-device adaptation can use gradient-free estimators (e.g., Simultaneous Perturbation Stochastic Approximation (SPSA)) ([Alexeev et al., 2025](#); [Yang et al., 2019](#)). Together, these provide quantitative criteria for calibration strategy: when $\sigma_\tau^2$ is high and $K$ is sufficient, adaptation outperforms a fixed controller.

## 2. Related Work

**Meta-learning.** Meta-learning algorithms such as model-agnostic meta-learning (MAML) ([Finn et al., 2017](#)) and its first-order variant (FOMAML) ([Nichol et al., 2018](#)) learn initializations enabling per-task adaptation. Theoretical analyses establish convergence guarantees under smoothness and PL conditions ([Karimi et al., 2016](#); [Fallah et al., 2020](#)) but do not address *when* adaptation outperforms non-adaptive policies or how gains scale with task variance. ([Deleu & Bengio, 2018](#)) show that adaptation can *decrease* performance when task variance is limited, highlighting the need for predictive criteria: the gap our scaling laws address.

**Task landscape conditions.** Recent analysis of MAML ([Collins et al., 2022](#)) characterizes when it outperforms non-adaptive learning, showing that gains require sufficient task diversity and geometric separation between optima. Our work builds on this by deriving quantitative scaling laws predicting the *magnitude* of adaptation gains as a function of task variance $\sigma_\tau^2$, where $\sigma_\tau^2$ corresponds to measurable device heterogeneity.

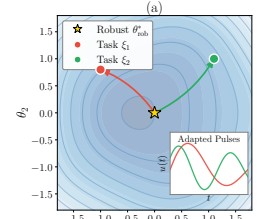 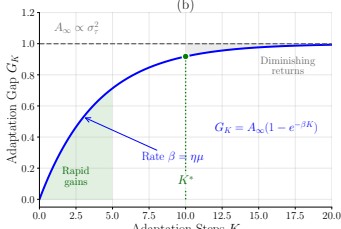

*Figure 2.* **Scaling law.** (a) In the loss landscape over control parameters $\theta_1$ and $\theta_2$, the robust (non-adaptive) controller ($\theta_{\text{rob}}^\star$, star) minimizes average loss but is suboptimal for individual devices (colored dots); gradient-based adaptation traces distinct trajectories toward device-specific optima, yielding different controls $u(t)$ (inset). (b) The adaptation gap follows $G_K = A_\infty(1 - e^{-\beta K})$, where $\beta = \eta\mu$ captures adaptation rate and $A_\infty \propto \sigma_\tau^2$ scales with task variance. $K^*$ marks diminishing returns.

**Quantum control.** GRAPE ([Khaneja et al., 2005](#)) optimizes pulses via differentiable dynamics but provides no finite-time adaptation guarantees; each new device requires re-optimization from scratch. RL approaches ([Bukov et al., 2018](#); [Niu et al., 2019](#); [Ernst et al., 2025](#)) learn device-specific policies but lack theoretical guarantees on when adaptation outperforms non-adaptive baselines. At the circuit level, AlphaTensor-Quantum ([Ruiz et al., 2025](#)) applies RL to minimize T-gate counts, demonstrating learned optimization can outperform hand-designed heuristics; however, this operates on discrete gate sequences rather than continuous pulses. Recently, Zhang *et al.* introduced metaQctrl ([Zhang et al., 2025](#)), achieving 99.99% fidelity under parameter uncertainty, but without scaling laws predicting *when* or *by how much* adaptation helps. Our work is complementary in that we derive explicit scaling laws that quantify these relationships as functions of gradient steps $K$, task variance $\sigma_\tau^2$, and landscape curvature $\mu$.

## 3. Problem Formulation

We now formalize the continuous-time control setting and define the *adaptation gap*.

### 3.1. System Dynamics and Control Objective

*Dynamical system.* We consider a family of differentiable control systems indexed by task parameter $\xi \in \mathbb{R}^n$:

$$\dot{x}(t) = f_\xi(x(t), u(t; \theta)), \quad (1)$$

where $x(t)$ is the system state, $u(t; \theta)$ is the control signal parameterized by $\theta$, and $f_\xi$ is a task-dependent dynamical linear operator. Each task $\xi$ specifies an environment instance drawn from distribution $\mathcal{P}(\xi)$. For example, device-specific relaxation time $T_1$ or device coupling $J$.

For classical control, $f_\xi$ is an ordinary differential equation; for open quantum systems, $x = \rho$ is the density matrix

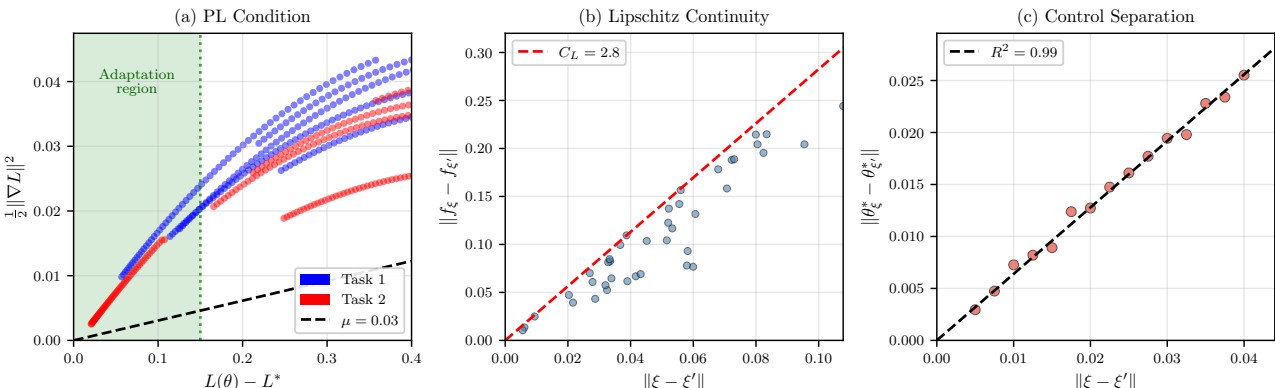

*Figure 3.* **Theory Validation** (a) PL condition: Optimization trajectories for two representative tasks (different colors) plotting gradient norm $\frac{1}{2}\|\nabla L\|^2$ versus optimality gap $L - L^*$. The PL inequality (Eq. 5) holds when points lie above the line with slope $\mu$. In the shaded *adaptation regime* (near the optimum, where $L - L^* < 0.14$) for optimum $L^\star$, the relationship is approximately linear with $\mu \approx 0.03$, confirming that gradient descent makes consistent progress. Outside this regime, the PL bound is violated as trajectories traverse saddle regions. (b) Lipschitz continuity: Lindbladian distance $\|f_\xi - f_{\xi'}\|$ versus task distance $\|\xi - \xi'\|$, where $\xi = (\Gamma_{\text{deph}}, \Gamma_{\text{relax}})$ are the dissipation rates defining each task. The linear bound $\|L_\xi - L_{\xi'}\| \leq C_L \|\xi - \xi'\|$ holds with $C_L \approx 2.8$, confirming Lemma 4.5. (c) Control separation: Distance between task-optimal controls $\|\theta_\xi^* - \theta_{\xi'}^*\|$ versus task parameter distance $\|\xi - \xi'\|$. The linear relationship ($R^2 = 0.98$) demonstrates consistency with Lemma 4.7 (different tasks, different optimal controls).

and $f_\xi$ corresponds to the Lindblad superoperator $\mathcal{L}_\xi$ (Manzano, 2020) governing dissipative evolution, with $u(t; \theta)$ specifying laser or microwave pulse waveforms.

*Loss function.* For a fixed task $\xi$, we measure performance via a final-time loss

$$L_\xi(\theta) = \ell(x_T(\theta, \xi), x_T^*), \tag{2}$$

where $x_T$ is the final state after evolution over a total duration $T$ and $\ell$ measures deviation from target $x_T^*$. In quantum control, we use the infidelity $\ell = 1 - \mathcal{F}$, where $\mathcal{F}$ is the state fidelity between states $\rho_1$ and $\rho_2$ (corresponding to the ideal final state and the measured final state from the gate operation) such that $\mathcal{F}(\rho_1, \rho_2) = \left(\text{Tr}\sqrt{\sqrt{\rho_1}\rho_2\sqrt{\rho_1}}\right)^2$.

### 3.2. Adaptive strategies

*Meta-learned adaptation.* Meta-learning (Finn et al., 2017) instead learns an initialization $\theta_0$ optimized for fast adaptation: rather than minimizing average loss directly, it minimizes post-adaptation loss. Given task $\xi$, the adapted policy $\pi_{\theta_K}$ is obtained via $K$ gradient steps according to

$$\theta_K(\xi) = \theta_0 - \eta \sum_{k=1}^{K} \nabla_\theta L_\xi(\theta_{k-1}). \tag{3}$$

*Baseline choice.* We use the meta-initialization $\theta_0$ as our baseline for computing the adaptation gap, measuring the benefit of task-specific adaptation beyond the learned initialization. We also compare against a fixed-average baseline and GRAPE (Khaneja et al., 2005) in Section 5 and Appendix A.5 to isolate the contributions of meta-learning and to benchmark against standard quantum optimal control.

### 3.3. The Adaptation Gap

We quantify the expected benefit of adaptation as

$$G_K = \mathbb{E}_{\xi \sim \mathcal{P}} \left[ L_\xi(\theta_0) - L_\xi(\theta_K(\xi)) \right], \tag{4}$$

where $\theta_0$ is the meta-initialization and $\theta_K(\xi)$ is the adapted policy after $K$ gradient steps on task $\xi$ drawn from distribution $\mathcal{P}$ (e.g., where $\mathcal{P}$ could correspond to the distribution of noise measurements measured over a batch of qubits). A positive $G_K$ indicates that task-specific adaptation provides measurable improvement over deploying the initialization directly. This metric answers whether $K$ gradient steps justify their computational overhead.

## 4. Theoretical Framework

We develop a framework characterizing when adaptation outperforms robust, non-adaptive optimization, combining rigorous results from quantum control theory (Lemmas 4.4–4.7) with scaling predictions (Figure 3).

### 4.1. Assumptions

We require structural conditions on the system, task distribution, and operating regime.

**Assumption 4.1** (System Structure). *The system satisfies:*

*(a) Controllability:* $\text{Lie}\{H_0, H_1, \ldots, H_m\} = \mathfrak{su}(d)$, *where $H_0$ is the static Hamiltonian and $\{H_1, \ldots, H_m\}$ are control Hamiltonians.*

*(b) Bounded controls:* $|u_k(t)| \leq U_{\max}$ *for all $k$ and $t$, reflecting hardware amplitude limits, where $U_{\max}$ is*

*the maximum control amplitude (e.g., laser power).*

These conditions are standard in quantum control (D'Alessandro, 2007). Controllability ensures all target states are reachable and bounded controls reflect hardware limitations.

**Assumption 4.2** (Task Structure). *Tasks $\xi = (\Gamma_{\text{deph}}, \Gamma_{\text{relax}}) \sim \mathcal{P}$ have bounded dissipation rates: $0 < \Gamma_{\min} \leq \Gamma_j(\xi) \leq \Gamma_{\max}$ for all noise channels $j$. More generally, $\xi$ could parameterize coupling strengths or other hardware variations; we focus on dissipation rates as these dominate calibration drift in superconducting qubits.*

This ensures the meta-learning problem is well-posed; dissipation rates are neither zero nor unbounded.

**Assumption 4.3** (Operating Regime). *The loss landscape $\nabla_\theta L_\xi(\theta)$ satisfies the Polyak-Łojasiewicz (PL) condition locally near task-specific optima: for each task $\xi$, there exists $\mu(\xi) > 0$ such that*

$$\frac{1}{2}\|\nabla_\theta L_\xi(\theta)\|^2 \geq \mu(\xi)\big(L_\xi(\theta) - L_\xi^*\big) \tag{5}$$

*in a neighborhood of the optimum $\theta_\xi^*$.*

The constant $\mu(\xi)$ characterizes landscape sharpness for each task. For closed quantum systems, controllability ensures trap-free landscapes (Russell et al., 2017). The trap-free structure does not imply PL but suggests favorable geometry, and we verify PL empirically: aggregating across the task distribution, satisfaction is essentially universal at moderate thresholds for ID and remains high under strong OOD ($10\times$); even when the strict threshold is violated, the exponential fit on $G_K$ retains $R^2 > 0.95$ (Figure A.7, Appendix A.7). The qualitative scaling law form thus persists outside the strict PL basin and degrades gracefully; this is consistent with the KL relaxation discussed below.

The PL condition is empirically verified for the loss functions considered here, but it is worth noting that it places a strong constraint on loss function behavior. For the strongest form of the main result (an exponentially decaying gap bound), the PL condition is required. However, weaker but still useful results can be obtained for loss surfaces that admit the more general KL condition (Attouch et al., 2011), of which PL is a special case. In particular in the case of a KL loss function, one obtains a polynomial decaying gap of order $\mathcal{O}(k^{-1/(2\alpha-1)})$ instead of an exponentially decaying gap. The polynomial rate of convergence is controlled by the exponent $\alpha$ in the KL condition, which in turn expresses how slowly the gradient magnitude grows near the optimum. This fallback criterion provides theoretical guarantees even in the case where the strict PL condition is violated in favor of the weaker KL condition. The proof of the main results in the appendix are derived under this more general condition, with the PL case recovered as an outcome.

Figure 3(a) confirms that optimization trajectories in our open-system setting satisfy Eq. (5) with $\mu \approx 0.03$ in the adaptation regime (shaded linear region near the optimum where gradient descent makes consistent progress).

## 4.2. Optimization Landscape

Three results characterize the optimization geometry.

**Lemma 4.4** (Trap-Free Landscape). *Under Assumption 4.1, for closed quantum systems the loss (fidelity) landscape contains no suboptimal local minima (maxima).*

*Proof.* See Appendix C.1 for a proof. □

This result follows from the controllability rank condition: local surjectivity of the endpoint map prevents suboptimal traps. See (Russell et al., 2017) and Appendix C.1.

**Lemma 4.5** (Lipschitz Task Dependence). *Under Assumptions 4.1 and 4.2, the quantum dynamics operators $f_\xi$ satisfy*

$$\|f_\xi - f_{\xi'}\| \leq C_L \|\xi - \xi'\| \tag{6}$$

*where $C_L$ depends on task-specific bounds.*

*Proof.* See Appendix C.1 for the proof. □

This ensures the meta-learning problem is well-posed: nearby tasks induce nearby dynamics. Figure 3(b) confirms this bound, showing dynamics operator (Lindbladian) distance is bounded by $C_L \approx 2.8$ times the task distance.

## 4.3. Task Variance and Control Separation

A central question is whether different tasks require meaningfully different optimal controls. We quantify this here:

**Definition 4.6** (Task Variance). *For a task distribution $\mathcal{P}$ over task parameters $\xi$, we define the task variance as*

$$\sigma_\tau^2 = \sum_i \text{Var}_{\xi \sim \mathcal{P}}[\xi_i]. \tag{7}$$

*In our quantum control experiments, $\xi = (\Gamma_{\text{deph}}, \Gamma_{\text{relax}})$, where $\Gamma_{\text{deph}} = \frac{1}{T_2}$ (decoherence) and $\Gamma_{\text{relax}} = \frac{1}{T_1}$ (relaxation). So $\sigma_\tau^2 = \text{Var}(\Gamma_{\text{deph}}) + \text{Var}(\Gamma_{\text{relax}})$.*

The following key theoretical result connects task variance to control separation:

**Lemma 4.7** (Control Separation). *Let $\theta^*(\xi)$ denote the optimal control for task $\xi \in \Xi$. We will fix a reference task $\xi^* \in \Xi$ that has an optimum $\theta^\star = \theta^\star(\xi^\star)$ and assume the following conditions:*

1. **Smoothness:** *$L_\xi(\theta)$ is twice continuously differentiable in $(\theta, \xi)$.*

2. **Strong convexity at the optimum:** *The Hessian* $H = \nabla^2_{\theta\theta} L_{\xi^\star}(\theta^\star)$ *satisfies* $H \succeq \mu I$ *for constant* $\mu > 0$.

3. **Task sensitivity:** *The mixed partial derivative* $M = \nabla^2_{\xi\theta} L_{\xi^\star}(\theta^\star)$ *satisfies that* $\sigma_{\min}(M) > 0$ *(with* $\sigma_{\min}(M)$ *as the smallest singular value of* $M$*).*

4. **Sufficient control dimension:** $\dim(\theta) \geq \dim(\xi)$.

*Then there exists a neighborhood* $\mathcal{B}$ *of* $\xi^\star$ *and constant* $C_{\mathrm{sep}} > 0$ *such that for all* $\xi \in \mathcal{B}$:

$$\|\theta^*(\xi) - \theta^*(\xi^*)\| \geq C_{\mathrm{sep}}\|\xi - \xi^*\| \qquad (8)$$

*such that* $C_{\mathrm{sep}} = \sigma_{\min}(H^{-1}M) \geq \sigma_{\min}(M)/\|H\| > 0$.

*Proof.* See Appendix C.1 for the proof. □

This lemma ensures that task variance translates to meaningful differences in optimal controls. We validate this empirically in Figure 3(c) ($R^2 = 0.98$).

## 4.4. Main Result

We now state our main theorem, which characterizes how the adaptation gap scales with system properties.

**Theorem 4.8** (Adaptation Gap Scaling Law)**.** *Consider a system with task variance* $\sigma_\tau^2$ *(in the quantum case, Definition 4.6) satisfying Assumptions 4.1–4.3 and worst-case PL constant* $\mu_{\min}$. *Suppose the meta-initialization* $\theta_0$ *lies within the basin of attraction where the PL condition (Assumption 4.3) holds. Then the expected adaptation gap after* $K$ *gradient steps with learning rate* $\eta$ *satisfies*

$$G_K \geq A_\infty \left(1 - e^{-\beta K}\right) \qquad (9)$$

*where* $A_\infty = c\,\sigma_\tau^2$ *is the asymptotic gap,* $\beta = \eta\mu_{\min}$ *is the adaptation rate, and* $c > 0$ *aggregates system constants. Here* $\mu_{\min} = \inf_\xi \mu(\xi)$ *is the worst-case PL constant.*

*Remark (proof roadmap):* The scaling law decomposes into three components (full proof in Appendix C.2):

(a) **Functional form** $G_K \propto (1 - e^{-\beta K})$: rigorously proven under Assumption 4.3 (local PL); see Parts A–C. The proof of the decay rate of the gap follows the same form as standard arguments for the convergence of first-order gradient descent techniques.

(b) **Rate** $\beta = \eta\mu_{\min}$: rigorous given PL plus a standard L-smoothness regularity condition and a bounded learning rate $\eta \leq 1/L$; also Parts A–C.

(c) **Ceiling** $A_\infty \propto \sigma_\tau^2$: relies on four additional local assumptions: (i) quadratic loss approximation near optima, (ii) approximately task-independent curvature

$H$, (iii) linear optima map $\theta^*(\xi) = A\xi + b$, and (iv) isotropic task variance. Under these assumptions the MAML initialization coincides with the average-task solution, and the expected initial suboptimality reduces to a trace-covariance expression proportional to $\sigma_\tau^2$.

Components (a) and (b) hold robustly; (c) is a local approximation that may pick up higher-order corrections under large distribution shift.

*Proof.* See Appendix C.2 for the proof. □

## 4.5. Practical Implications

The corollaries below depend only on components (a) and (b) of the remark above; they remain actionable even when the ceiling decomposition (c) is approximate.

**Corollary 4.9** (Diminishing Returns Threshold)**.** *To capture fraction* $\alpha$ *of the maximum adaptation gain* $A_\infty$, *the required gradient budget is*

$$K_\alpha = \frac{1}{\beta} \log \frac{1}{1 - \alpha}. \qquad (10)$$

*(e.g.,* $K_{0.95} = 3/\beta$ *captures 95% of the asymptotic gain).*

**Corollary 4.10** (When non-adaptive deployment suffices)**.** *Adaptation provides negligible benefit when:*

(a) $\sigma_\tau^2$ *is small (tasks are similar), or*

(b) $\beta \ll 1/K_{\mathrm{budget}}$ *(adaptation is slow).*

*In these regimes, non-adaptive deployment achieves comparable performance with lower overhead.*

# 5. Quantum Control Experiments

We validate Section 4 results here. Full experimental details, algorithms, and expanded results appear in Appendix A and Appendix B. A quantum control primer is in Appendix D.

## 5.1. Experimental Setup

To test whether the scaling law holds under realistic quantum noise, we construct a differentiable simulator capturing the essential physics of qubit calibration.

**System model.** We model task variations in qubit dynamics by solving the Lindblad master equation for the density matrix (quantum state) $\rho$ (a specific case of Eq. (1)), in which

$$\dot{\rho}(t) = -i[H_0 + \textstyle\sum_k u_k(t)H_k, \rho] + \textstyle\sum_j \Gamma_j(\xi)\mathcal{D}[L_j]\rho, \quad (11)$$

where $\Gamma_j(\xi)$ are task-dependent noise rates. We have that $\xi = (\Gamma_{\mathrm{deph}}, \Gamma_{\mathrm{relax}})$ directly, so the task parameter is the noise

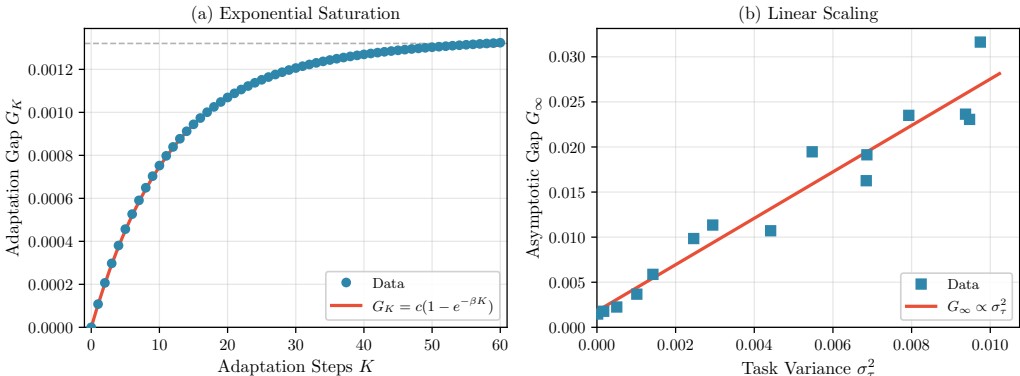

*Figure 4.* **Scaling law validation.** (a) Adaptation gap $G_K$ versus inner-loop steps $K$ exhibits exponential saturation ($R^2 > 0.999$) for small task variance (in-distribution), consistent with Theorem 4.8. The fitted curve $G_K = c(1 - e^{-\beta K})$ captures the transition from rapid gains to diminishing returns. (b) Asymptotic gap $G_\infty$ scales linearly with actual task variance $\sigma_\tau^2 = \mathrm{Var}(\Gamma_{\mathrm{deph}}) + \mathrm{Var}(\Gamma_{\mathrm{relax}})$ ($R^2 = 0.94$), evaluated across $\sim 0.01$–$5\times$ the nominal training variance. This is consistent with the predicted proportionality $A_\infty \propto \sigma_\tau^2$.

rate tuple itself, with jump operators $L_j$ parameterizing relaxation and dephasing, where $\mathcal{D}[L]\rho = L\rho L^\dagger - \frac{1}{2}\{L^\dagger L, \rho\}$ is the dissipator superoperator. Control pulses $u_k(t)$ weigh the control Hamiltonians $H_k$, competing against noise channels that scramble the quantum state. See Appendix D for system details. Tasks are sampled from $\mathcal{P}(\xi)$ with variance $\sigma_\tau^2 = \mathrm{Var}(\Gamma_{\mathrm{deph}}) + \mathrm{Var}(\Gamma_{\mathrm{relax}})$ (Figure 1(a)).

**Policy and training.** The control policy $\pi_\theta$ maps task features to piecewise-constant control amplitudes via a two-layer multi-layer perceptron (MLP). Following standard meta-learning practice (Finn et al., 2017), the policy receives task parameters $\xi = (\Gamma_{\mathrm{deph}}, \Gamma_{\mathrm{relax}})$ as context input, analogous to task embeddings. This differs from the theoretical formulation where $u(t; \theta)$ depends only on $\theta$; in practice, the $\xi$-conditioning enables task-appropriate pulse initialization, while the adaptation gap $G_K$ still measures improvement from gradient updates to $\theta$ itself. In deployment , tasks $\xi$ such as noise rates can be estimated via standard $T_1/T_2$ characterization protocols that already form part of any calibration workflow (Krantz et al., 2019). See Appendix A and B for details on training.

**Baselines and metrics.** We compare three strategies: (1) meta-initialization $\theta_0$ (FOMAML, before adaptation), (2) fixed-average baseline (trained on mean task $\bar{\xi}$), and (3) GRAPE optimized per-task from scratch (results in Appendix A.5). We measure the adaptation gap $\mathbb{E}_\xi[L_\xi(\theta_0) - L_\xi(\theta_K(\xi))]$ and fit to $c(1 - e^{-\beta K})$ to extract adaptation rate $\beta$ and asymptote $A_\infty = c$.

### 5.2. Single-Qubit Gate Calibration

The X gate (a single-qubit bit-flip) under dephasing and relaxation captures the essential calibration challenge: adapting to device-specific decoherence that drifts over time.

Figure 4(a) validates Theorem 4.8's prediction that $G_K$ saturates exponentially. The adaptation gap fits $c(1 - e^{-\beta K})$ with $R^2 > 0.99$ and rate $\beta = 0.083$. Each gradient step captures roughly 8% of the remaining improvement; after $K \approx 20$ steps, diminishing returns set in.

**Linear scaling with task variance.** To vary $\sigma_\tau^2$ in Figure 4(b), we scale the uniform distribution support by a diversity factor $d \in [0.1, 3.0]$, with $\sigma_\tau^2 = \mathrm{Var}(\Gamma_{\mathrm{deph}}) + \mathrm{Var}(\Gamma_{\mathrm{relax}})$ computed empirically from sampled tasks. Figure 4(b) confirms that $G_\infty$ scales linearly with $\sigma_\tau^2$ ($R^2 = 0.94$), evaluated across $\sim 0.01$–$5\times$ the nominal training variance. Crucially, when task variance is small ($\sigma_\tau^2 < 0.002$), adaptation provides negligible measurable benefit ($G_\infty \approx 0$). This is precisely the regime where a fixed controller suffices.

**Adaptation dynamics.** Figure 5 compares initialization strategies on challenging tasks. Meta-initialization (FOMAML) starts at high fidelity ($\mathcal{F} = 0.959$) and improves only marginally to $\mathcal{F} = 0.960$ after $K = 10$ steps; the nearly indistinguishable pulse waveforms (Figure 5(c)) confirm that FOMAML already generalizes well within the task distribution, leaving little room for task-specific correction. The fixed-average baseline tells the opposite story: starting at $\mathcal{F} = 0.881$, it improves substantially to $\mathcal{F} = 0.935$, with visibly restructured pulses (orange). This contrast illustrates the scaling law's core prediction: when the initialization is already near task-specific optima, adaptation gains are minimal; when initialization is poor, adaptation provides large improvements.

**Ablation studies.** Appendix A.4 confirms that $\beta$ scales linearly with learning rate $\eta$ while $A_\infty$ remains independent of optimization hyperparameters.

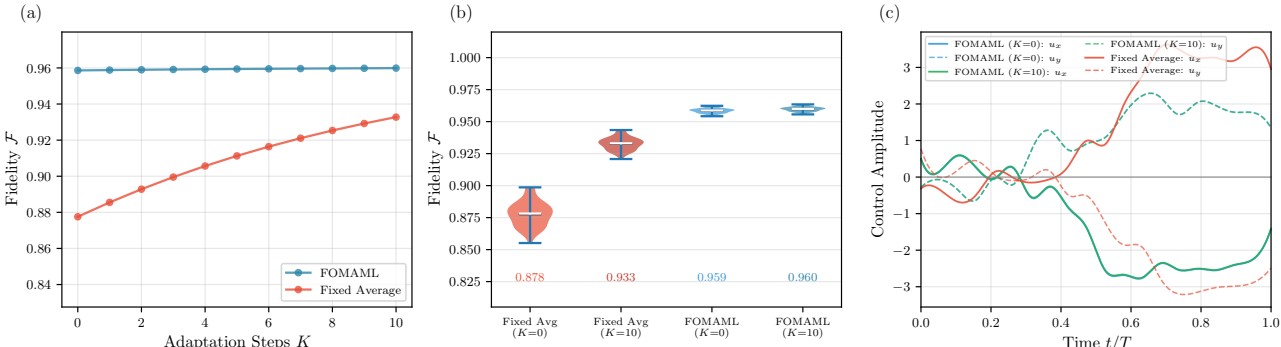

*Figure 5.* **Adaptation dynamics (mild out-of-distribution (OOD), $\sim 1.1\times$ training noise).** (a) Fidelity versus adaptation steps $K$ for two initialization strategies: meta-learned (FOMAML) starts higher and converges faster than the fixed-average baseline. (b) Fidelity distributions across strategies: fixed-average baseline achieves $\mathcal{F} = 0.881$ before adaptation and improves substantially to $\mathcal{F} = 0.935$ after $K = 10$ steps; meta-initialization achieves $\mathcal{F} = 0.959$ before adaptation, improving only marginally to $\mathcal{F} = 0.960$ after adaptation. (c) Learned pulse sequences: FOMAML pulses before and after adaptation are nearly indistinguishable, confirming the initialization already generalizes well. Fixed-average optimized pulses after $K = 10$ adaptation steps (orange) show qualitatively different structure.

## 5.3. Two-Qubit Entangling Gate

Two-qubit gates present the primary calibration bottleneck in current quantum processors. The CZ gate couples two qubits via a ZZ interaction, doubling the noise channels and tripling the control parameters.

**High-noise regime.** We deliberately stress-test by inflating noise rates ($\Gamma_{\text{deph}}, \Gamma_{\text{relax}}$) by $10\times$ above typical operating conditions, mimicking a device that has drifted badly out of calibration, or a newly fabricated chip before tune-up. Order-of-magnitude $T_1$ fluctuations documented on production superconducting processors (Carroll et al., 2022; Klimov et al., 2018) place the $10\times$ regime at one end of a continuous spectrum rather than as an isolated stress test. The two-qubit system has six control fields $(u_{x,1}, u_{y,1}, u_{z,1}, u_{x,2}, u_{y,2}, u_{z,2})$ and richer dynamics due to the ZZ coupling between qubits. Figure 6(c) confirms exponential saturation with $R^2 = 0.986$ and fitted rate $\beta = 0.333$, reflecting the richer control landscape.

**Fidelity improvement.** Meta-initialization yields $\mathcal{F}_0 = 54.2\%$ in this high-noise regime (Figure 6(a)), far below the fault-tolerance threshold. After $K = 10$ adaptation steps, task-specific pulse corrections achieve $\mathcal{F} = 95.7\%$ (Figure 6(b)), a 41.5% improvement.

**Hamiltonian parameter variation.** To test whether the scaling law extends beyond noise-rate variation, we conduct a second two-qubit experiment where the task parameter is the ZZ coupling strength $J$ rather than dissipation rates. This reflects realistic device heterogeneity where coupling strengths vary across qubit pairs due to fabrication tolerances and it also models modern tunable-coupler architectures where $J$ is dynamically adjustable (Yan et al.,

2018). See full details in Appendix A.6.

**Control waveform adaptation.** For decoherence-dominated tasks (Figure 6), adaptation under $10\times$ elevated noise transforms the meta-initialized pulses from $\mathcal{F} = 54.2\%$ to $\mathcal{F} = 95.7\%$. The most pronounced change is in $u_{z,2}$ (light blue): before adaptation (a), this channel remains essentially inactive; after adaptation (b), it exhibits oscillatory modulation at an elevated amplitude. This active single-qubit $z$-control (dynamic adjustment of qubit frequency), enabled experimentally via flux tuning or AC Stark shifts, effectively implements dynamical decoupling to refocus low-frequency dephasing noise that would otherwise destroy the entangling phase.

For coupling-dominated tasks (Figure A.6(c)–(f)), adaptation produces structurally distinct pulses for each coupling strength $J$. Strong native coupling ($J = 9.0$) requires large $u_{ZZ}$ amplitudes to compensate by accumulating sufficient entangling phase, while weak coupling ($J = 1.0$) demands slowly varying $u_{ZZ}$ waveforms.

## 5.4. Operational Decision Protocol

Algorithm 1 converts Theorem 4.8 into a pre-adaptation decision rule from a few probe inner-loop steps. For a 100-qubit processor with daily 30–90 minute calibration windows (IBM Quantum, 2025), applying the few-shot protocol per gate avoids both wasted compute on near-converged tasks ($\sigma_\tau^2$ small) and underprovisioned budgets on drifted devices ($\sigma_\tau^2$ large).

**Prediction accuracy.** With $N = 5$ probe steps, the relative error of $\hat{K}_{0.95}$ stays within 3–19% across $2\times$–$5\times$ OOD. See Appendix A.8.

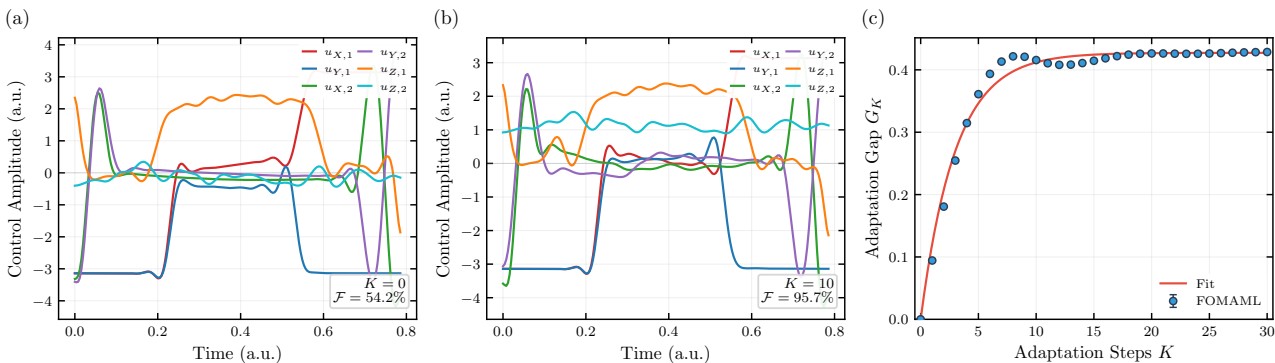

**Figure 6.** **Two-qubit CZ gate adaptation under high noise (strong OOD, $10\times$ training noise; consistent with documented worst-case $T_1$ fluctuations on superconducting hardware).** (a) Meta-initialized control pulses ($K = 0$) yield $\mathcal{F} = 54.2\%$ gate fidelity in a high-noise environment. (b) After $K = 10$ adaptation steps, task-specific corrections achieve $\mathcal{F} = 95.7\%$. (c) Adaptation gap $G_K$ follows the predicted exponential saturation $G_K = c(1 - e^{-\beta K})$ with $R^2 = 0.986$ and $\beta = 0.333$.

---

**Algorithm 1** Few-Shot Pre-Adaptation Decision Protocol

---
**Require:** $\theta_0$, probe count $N \in \{3, 4, 5\}$, target gain fraction $\alpha$ (e.g., 0.95), benefit threshold $\epsilon$, representative task $\xi$
1: Initialize $\theta^{(0)} \leftarrow \theta_0$
2: **for** $k = 1, \ldots, N$ **do**
3:     $\theta^{(k)} \leftarrow \theta^{(k-1)} - \eta \nabla_\theta L_\xi(\theta^{(k-1)})$       (inner-loop step)
4:     $G_k \leftarrow L_\xi(\theta_0) - L_\xi(\theta^{(k)})$          (adaptation gap)
5: **end for**
6: Fit $G_k \approx \hat{A}_\infty(1 - e^{-\hat{\beta}k})$ to $\{(k, G_k)\}_{k=1}^N$, record $R^2$
7: **PL diagnostic:** if $R^2 < 0.9$, flag departure from local PL
8: Compute budget $\hat{K}_\alpha = \frac{1}{\hat{\beta}} \log \frac{1}{1-\alpha}$       (Corollary 4.9)
9: **return** don't adapt if $\hat{A}_\infty < \epsilon$; else *adapt with budget* $\hat{K}_\alpha$

---

**Adaptation vs. conditioning.** Because $\pi_\theta$ takes $\xi$ as context (Section 5.1), one might worry the headline gains stem from conditioning rather than adaptation. An ablation crossing $\xi$-access (none / full / 30% noisy) with adaptation (none / $K{=}30$) shows adaptation alone yields $+0.184$ (mild OOD) and $+0.158$ (strong OOD, $10\times$) over a blind baseline, while noisy $\xi$ matches the full method (Figure A.9).

## 6. Discussion

**Interpretation.** The scaling law $G_K \geq A_\infty(1 - e^{-\beta K})$ separates ceiling from speed: $A_\infty \propto \sigma_\tau^2$ sets the maximum gain from adaptation (confirmed at $\sigma_\tau^2 < 0.002$ where the gap is small), while $\beta = \eta \mu_{\min}$ controls convergence rate (confirmed in Figure A.4 where $\beta \propto \eta$). The framework extends beyond quantum control to other systems with differentiable dynamics and favorable loss geometry. The linear quadratic regulator (LQR) validation in Appendix A.3 suggests the scaling law is not quantum-specific.

**Scaling with system complexity.** A notable feature of our results is the disparity in adaptation gap magnitude: $\sim$40% for CZ gates versus $\sim$0.1% for single-qubit X gates.

Three factors contribute: (i) distribution shift, since the two-qubit experiments use $10\times$ higher noise than training; (ii) constraint proliferation, with CZ gates optimizing over 12 input states versus two for X gates; and (iii) control capacity (6 control dimensions offer more directions).

**Limitations.** We address two types of limitations:
*(i) Theoretical Assumptions:* Our analysis relies on local PL near task optima. Under large distribution shift this can fail, manifesting as low $R^2$ of the fit or small $\mu$ (Figure A.4(b) at high $\eta$). Algorithm 1 flags this in deployment ($R^2 < 0.9$ on probe steps). The KL polynomial-decay form (Appendix C.2) provides a fallback guarantee.

*(ii) Simulation Requirement:* Our framework requires gradients through the dynamics, which is not feasible on hardware. More broadly, the simulate-then-deploy paradigm presumes the simulator captures the dominant noise channels: finite-shot noise, intra-episode drift, and imperfect knowledge of $\xi$ all introduce a simulation-to-hardware gap. These effects can be absorbed into an effective parameters.

## 7. Conclusion

From optimization geometry, we derived scaling laws predicting when adaptation outperforms non-adaptive baselines, and converted them into a few-shot decision protocol for adaptive quantum control. The law gives actionable guidance: adaptation wins when device variance is high and budget sufficient. For quantum scalable processors, this turns calibration from a fixed cost into a budgeted decision per gate. Extending these scaling laws to experimental quantum hardware systems is left for important future work.

## Impact Statement

This work bridges the gap between meta-learning theory and the operational realities of quantum computing. By deriving physics-informed scaling laws, we provide a blueprint for applying machine learning to control systems where hardware variation is the dominant constraint. Beyond quantum control, our framework for analyzing the "adaptation gap," linking task variance ($\sigma_\tau^2$) to meta-learning returns ($G_K$) via the PL condition, offers a generalizable template for evaluating meta-learning utility in other physical domains, such as robotics (sim-to-real transfer) and fusion reactor control.

Regarding broader impacts: quantum computing poses known risks, particularly to cryptographic infrastructure. However, this work addresses calibration efficiency rather than computational capability, and the scaling laws we derive are agnostic to the end application. The primary near-term effect is reducing energy consumption in calibration workflows. While our specific contribution targets calibration efficiency rather than computational capability, faster and cheaper calibration is a general-purpose enabler that lowers the cost of all downstream quantum applications, including those with cryptanalytic relevance. We see no near-term mechanism by which calibration speedups would unilaterally accelerate any specific harmful application, but acknowledge that calibration is a shared infrastructure layer.

## Acknowledgments

Approved for Public Release; Distribution Unlimited. Public Release Case Number 26-1084. ©2026 The MITRE Corporation. All rights reserved. This work is supported by the MITRE Independent Research and Development Program. We thank Taylor Patti (NVIDIA), Murphy Niu (UCSB), Victor Batista (Yale), and colleagues at NVIDIA for fruitful discussions.

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

# Appendix

## Appendix A: Additional Experimental Results

We provide additional experimental results that complement the main text. Our experiments focus on single-qubit and two-qubit gates rather than many-body control problems. This scope is deliberate: universal quantum computation factorizes into these primitive operations, and practical calibration workflows tune gates individually or pairwise. The following sections provide training diagnostics, hyperparameters, ablation studies, baseline comparisons, and extended validation on classical control and tunable-coupler architectures.

### A.1: Training Dynamics

We first examine the stability and convergence properties of the meta-training procedure. Figure A.1 presents four diagnostic metrics tracked over 2,000 meta-iterations.

Several observations confirm healthy meta-training dynamics. First, the training loss decreases monotonically over approximately two orders of magnitude, with no evidence of instability or divergence (panel a). Second, the validation metrics stabilize after roughly 500 iterations: both pre- and post-adaptation losses plateau, and the adaptation gap converges.

This gap magnitude aligns with our scaling law predictions given the task variance used in training. Third, the gradient norm decays smoothly without explosions or oscillations (panel c).

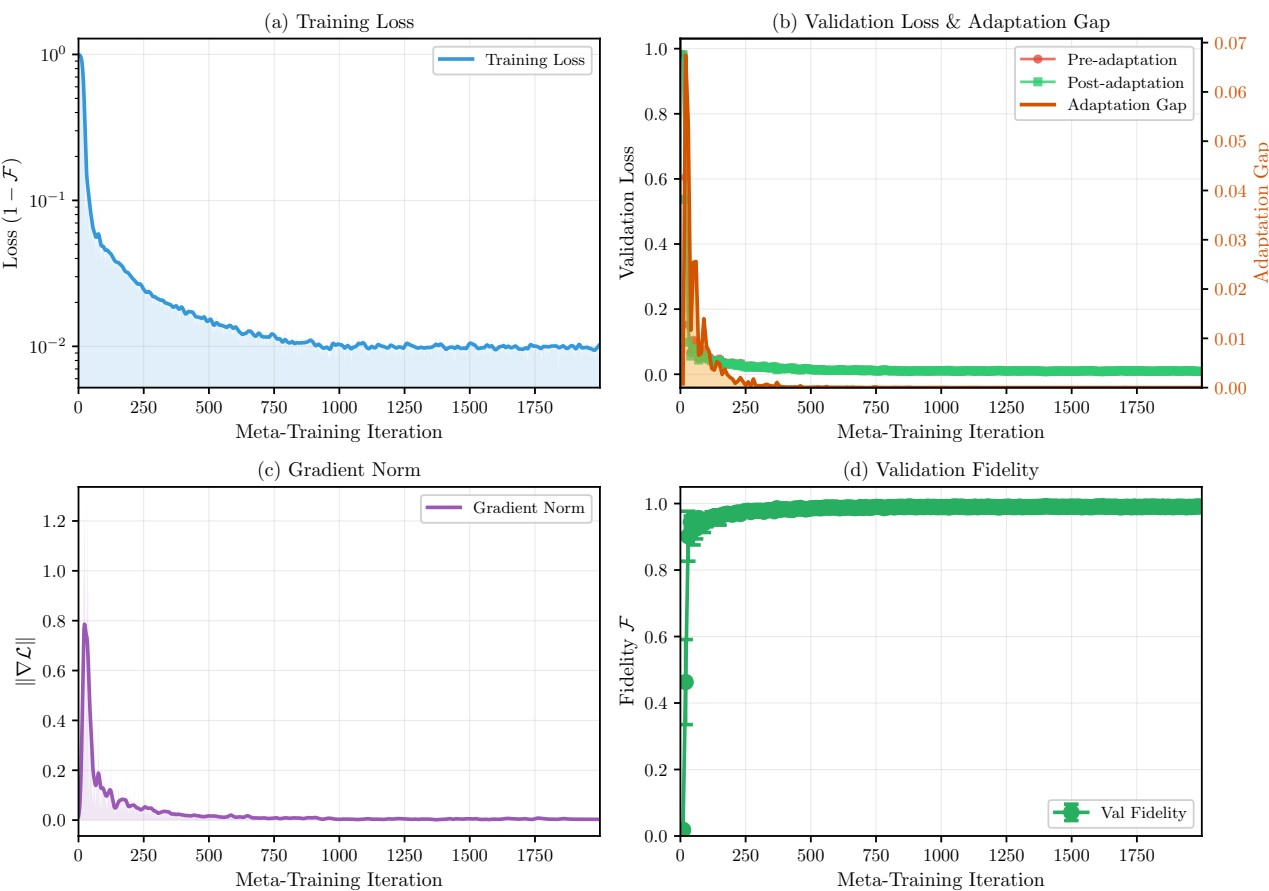

*Figure A.1.* **Meta-training dynamics (X gate).** (a) Training loss decreases over two orders of magnitude with stable convergence. (b) Validation loss (pre-and post-adaptation) and adaptation gap stabilize after ~500 iterations, consistent with scaling law predictions. (c) Gradient norm decays smoothly without explosions, indicating stable optimization. (d) Validation fidelity reaches $\mathcal{F} > 0.98$ with low variance across tasks, confirming the meta-initialization generalizes well.

## A.2: Hyperparameters and Fitting Parameters

*Table A.1.* Experimental settings and fitted scaling law parameters.

| Category | Parameter | Single-Qubit (X) [Figure 4] | Two-Qubit (CZ) [Figure 6] | LQR |
|---|---|---|---|---|
| Training Distribution | $\Gamma_{\text{deph}}$ range | $[0.02, 0.15]$ | $[0.0001, 0.001]$ | — |
| | $\Gamma_{\text{relax}}$ range | $[0.01, 0.08]$ | $[5 \times 10^{-5}, 0.0005]$ | — |
| | Distribution | Uniform | Uniform | Gaussian |
| Adaptation Distribution | $\Gamma_{\text{deph}}$ range | $[0.02, 0.15]$ | $[0.001, 0.01]$ | — |
| | $\Gamma_{\text{relax}}$ range | $[0.01, 0.08]$ | $[0.0005, 0.005]$ | — |
| | Distribution | Uniform | Uniform | Gaussian |
| | OOD ($10\times$ noise) | No | Yes | No |
| Scaling Law Fit | $c$ (asymptote) | 0.00132 | 0.4273 | 0.0029 |
| | $\beta$ (rate) | 0.0834 | 0.333 | 0.042 |
| | $R^2$ | 0.9998 | 0.9856 | 0.999 |

*Table A.2.* Meta-training hyperparameters for single-qubit and two-qubit experiments.

| Hyperparameter | Single-Qubit (X Gate) | Two-Qubit (CZ Gate) |
|---|---|---|
| *Policy Architecture* | | |
| Task features | $\left(\frac{\Gamma_{\text{deph}}}{0.1}, \frac{\Gamma_{\text{relax}}}{0.05}, \frac{\Gamma_{\text{deph}}+\Gamma_{\text{relax}}}{0.15}\right)$ | $\left(\frac{\Gamma_{\text{deph}}^i}{0.1}, \frac{\Gamma_{\text{relax}}^i}{0.05}\right) (i = 1, 2)$ |
| Task feature dimension | 3 | 4 |
| Hidden dimension | 128 | 256 |
| Hidden layers | 2 | 4 |
| Control segments | 60 | 30 |
| Number of controls | $2\ (u_x, u_y)$ | $6\ (u_{x,1}, u_{y,1}, u_{x,2}, u_{y,2}, u_{z,1}, u_{z,2})$ |
| Activation | Tanh | Tanh |
| Output scale | 1.0 | $\pi$ |
| *MAML Training* | | |
| Meta-iterations | 2000 | 2000 |
| Tasks per batch | 32 | 4 |
| Inner-loop steps | 5 | 3 |
| Inner-loop learning rate ($\eta_{\text{in}}$) | 0.01 | 0.05 |
| Meta learning rate ($\eta_{\text{out}}$) | 0.001 | 0.001 |
| Optimizer (Outer Loop) | Adam | AdamW |
| Optimizer (Inner Loop) | SGD | SGD |
| Weight decay | 0 | $10^{-4}$ |
| LR scheduler | None | Cosine annealing |
| Gradient clipping | 1.0 | 1.0 |
| *Simulation* | | |
| Integration method | RK4 | RK4 |
| Integration timestep ($\Delta t$) | 0.005 | 0.01 |
| Gate time ($T$) | 1.0 | $\pi/4 \approx 0.785$ |

*Table A.3.* Summary of adaptation regimes across experiments.

| Figure | Adaptation Regime | OOD Factor | Purpose |
|--------|-------------------|------------|---------|
| Fig. 4a | In-distribution | $1\times$ | Validate exponential saturation |
| Fig. 4b | Variable | $0.01$–$5\times$ | Validate $A_\infty \propto \sigma_\tau^2$ scaling |
| Fig. 5 | Mild OOD | $\sim 1.1\times$ | Compare initialization strategies |
| Fig. 6 | Strong OOD | $10\times$ | Stress-test adaptation limits |

## A.3: Classical Example

To demonstrate that our scaling law arises from optimization geometry rather than quantum-specific physics, we validate the framework on a classical linear-quadratic regulator (LQR) problem. This serves as a sanity check: if the exponential-linear structure emerges in a simple classical system with known analytical properties, it strengthens the claim that our results apply broadly to differentiable control.

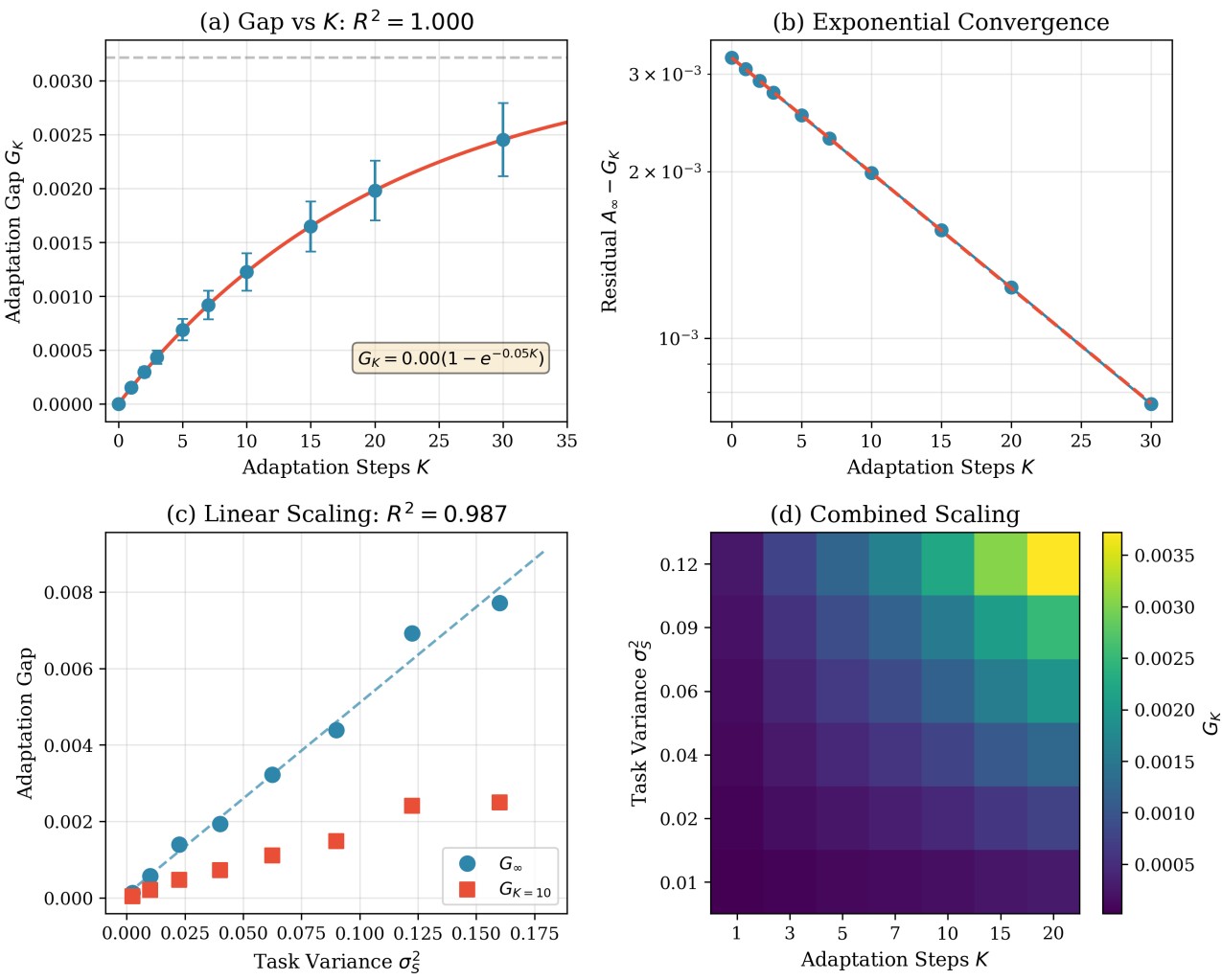

*Figure A.2.* **Classical LQR validation.** (a) Adaptation gap versus $K$ achieves $R^2 > 0.99$, demonstrating near-perfect exponential saturation on the mass-spring-damper system. (b) Log-scale residual $A_\infty - G_K$ confirms exponential convergence. (c) Linear scaling of $G_\infty$ with task variance ($R^2 = 0.987$); both asymptotic gap (blue) and finite-$K$ gap (red, $K$=10) follow the predicted relationship. (d) Combined scaling law: heatmap shows adaptation gap increases with both $K$ and $\sigma_\tau^2$, confirming the full structure $G_K \propto \sigma_\tau^2(1 - e^{-\beta K})$.

**System.** We consider a mass-spring-damper system $m\ddot{x} + c\dot{x} + kx = u$ where damping $c = 0.5$ and stiffness $k = 2.0$ are fixed, while mass $m \sim \mathcal{N}(1.0, \sigma_m^2)$ varies across tasks. The controller minimizes the infinite-horizon cost $J = \int_0^\infty (x^\top Q x + u^\top R u) \, dt$ with $Q = \mathrm{diag}(1.0, 0.1)$ and $R = 0.1$.

**Setup.** The non-adaptive baseline solves the Riccati equation (Uddin et al., 2019) for the mean system ($m = 1.0$), yielding a fixed gain $K_{\mathrm{rob}}$. For each task, adaptation performs gradient descent on the LQR cost starting from $K_{\mathrm{rob}}$.

The LQR results in Figure A.2 provide confirmation of the scaling law's generality.

One methodological difference from the quantum experiments merits discussion. Here, the non-adaptive baseline (the Riccati solution for the mean system) serves directly as the initialization for adaptation, rather than a separately meta-learned policy. This isolates the role of the PL condition and control separation in driving the scaling behavior, abstracting away meta-training dynamics. The fact that the same functional form emerges validates that our scaling law captures fundamental optimization geometry rather than artifacts of the MAML training procedure.

### A.4: Ablations

**Task variance to loss variance connection.**    A key step in our theoretical development connects physical task variance $\sigma_\tau^2$ to loss landscape geometry $\sigma_L^2$. Theorem 4.8 assumes these quantities scale proportionally. If task parameters vary, the optimal losses for different tasks should also vary, creating the suboptimality that adaptation can exploit. We validate this assumption empirically in Figure A.3.

The strong linear relationship ($R^2 = 0.987$) confirms that task parameter variance reliably predicts optimal loss variance across the range of noise conditions studied.

This empirical validation is important because the $\sigma_\tau^2 \to \sigma_L^2$ connection is invoked in the proof of the main theorem (Appendix C, Part A) but is difficult to establish rigorously without restrictive assumptions on the task distribution and loss landscape geometry. The strong empirical correlation ($R^2 = 0.987$) justifies our use of $\sigma_\tau^2$ as the independent variable in Figure 4(b) of the main text.

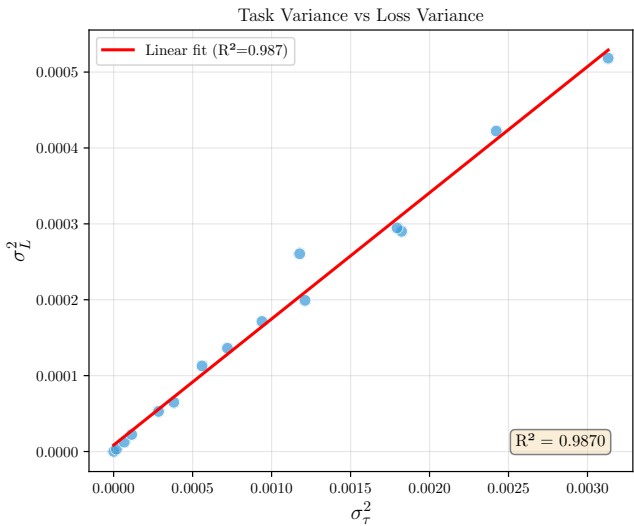

*Figure A.3.* **Empirical validation of the $\sigma_\tau^2 \to \sigma_L^2$ connection.** Task parameter variance $\sigma_\tau^2$ predicts optimal loss variance $\sigma_L^2$ with $R^2 = 0.987$, confirming the linear relationship assumed in Theorem 4.8 that connects physical task variance to loss landscape geometry.

**Learning rate sensitivity.** We investigate how the inner-loop learning rate $\eta$ affects adaptation dynamics. Our theory predicts that $\beta = \eta\mu_{\min}$, implying the adaptation rate should scale linearly with learning rate while the asymptotic gap $A_\infty$ remains independent of $\eta$ (since $A_\infty$ depends only on task variance $\sigma_\tau^2$). Figure A.4 tests these predictions.

Panel (a) confirms the predicted behavior: higher learning rates (yellow curves) achieve faster saturation while all curves converge to similar asymptotic gaps. This separation between rate and ceiling is a key prediction of Theorem 4.8. The learning rate controls how quickly you approach the ceiling, but the ceiling itself is set by task variance.

Panel (b) quantifies the relationship between $\eta$ and the fitted adaptation rate $\beta$. For $\eta \leq 2 \times 10^{-2}$, we observe approximately linear scaling, consistent with $\beta = \eta\mu_{\min}$. At higher learning rates, $\beta$ begins to plateau and eventually decrease. This departure from linearity indicates that gradient descent overshoots the basin where the local PL condition holds.

The rollover point provides practical guidance: learning rates beyond this threshold waste computation without accelerating adaptation.

These ablations have practical implications for hyperparameter selection. If computational budget limits the number of inner-loop steps $K$, practitioners should increase $\eta$ up to the linear regime boundary to maximize adaptation speed. Conversely, if stability is paramount (e.g., on real hardware with noisy gradients), lower learning rates provide more reliable convergence at the cost of requiring more adaptation steps.

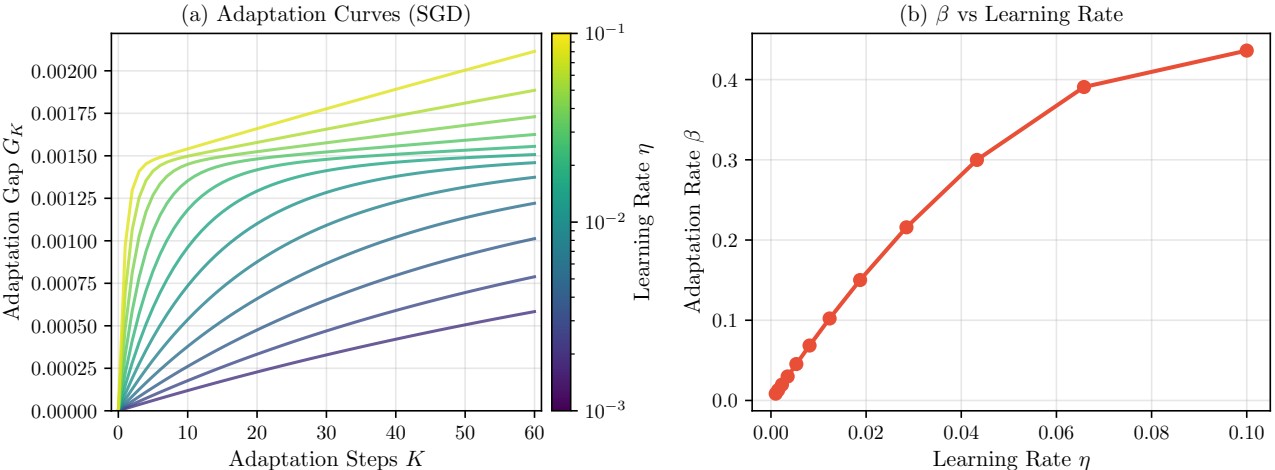

*Figure A.4.* **Learning rate sensitivity.** (a) Adaptation curves for different inner-loop learning rates $\eta$: higher $\eta$ (yellow) yields faster saturation while lower $\eta$ (purple) requires more steps. All curves converge to similar asymptotic gaps $A_\infty$, confirming that $A_\infty$ is independent of learning rate as predicted. (b) Fitted adaptation rate $\beta$ scales approximately linearly with $\eta$ for $\eta \leq 2 \times 10^{-2}$; the decrease at higher learning rates indicates departure from the local PL regime where gradient descent overshoots.

## A.5: Comparison to GRAPE Baseline

We compare our meta-learning approach against GRAPE (Gradient Ascent Pulse Engineering), the standard method for quantum optimal control (Khaneja et al., 2005). This comparison addresses whether the benefits of meta-learned adaptation justify its overhead relative to established pulse optimization techniques.

We consider four strategies: (1) *non-adaptive GRAPE*, which optimizes a single pulse sequence for the mean task distribution; (2) *per-task GRAPE*, which optimizes from scratch for each individual task; (3) *FOMAML (K=0)*, the meta-learned initialization before any task-specific adaptation; and (4) *FOMAML (K=10)*, the meta-learned policy after 10 gradient adaptation steps.

Figure A.5(a) shows that per-task GRAPE achieves the highest gate fidelity (99.3%), but requires warm-starting from a non-adaptive baseline and 200 optimization steps per task. In contrast, FOMAML's meta-initialization alone (K=0) achieves 98.9% fidelity with zero per-task optimization for tasks within the training distribution, and additional adaptation steps (K=10) provide negligible improvement.

Figure A.5(b) reveals the practical advantage: to match FOMAML's zero-shot performance of 98.9%, GRAPE from scratch requires over 150 iterations and still falls short, reaching only 97.5%. This gap reflects a fundamental limitation: GRAPE optimizes pulses for a single task instance, whereas FOMAML learns an initialization that generalizes across the task distribution. Consequently, FOMAML can be deployed immediately on new tasks without per-task optimization, while GRAPE must re-optimize from scratch for each new noise configuration. This efficiency gain is critical for addressing the calibration bottleneck, where frequent recalibration demands rapid deployment rather than lengthy re-optimization.

These results are consistent with our theoretical framework. Per Corollary 4.9, adaptation outperforms non-adaptive models when task variance $\sigma_\tau^2$ is non-negligible and sufficient adaptation steps $K > K^*$ are taken.

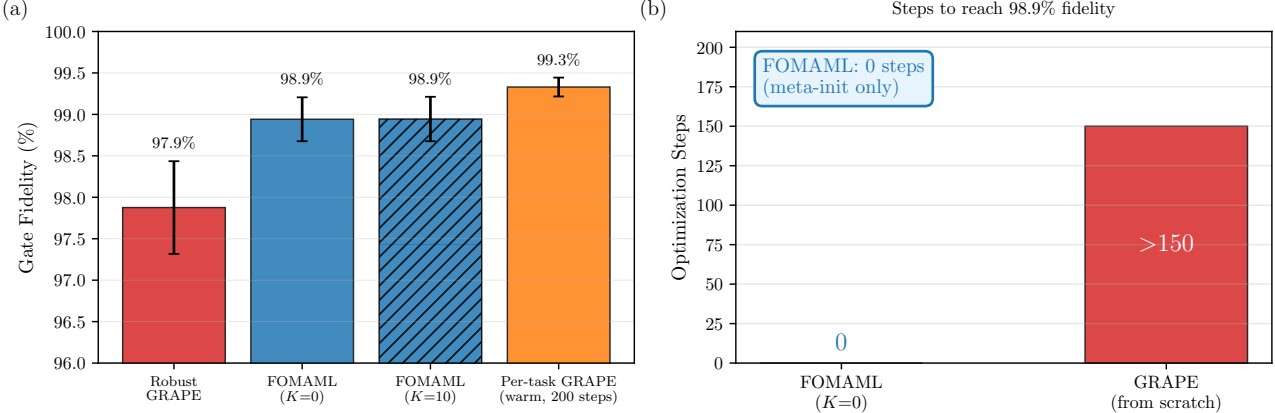

*Figure A.5.* **Comparison to GRAPE baseline.** (a) Gate fidelity across control strategies: non-adaptive GRAPE optimized for the mean task distribution (97.9%), FOMAML meta-initialization with zero adaptation steps (98.9%), FOMAML after $K$=10 adaptation steps (98.9%), and per-task GRAPE warm-started with 200 optimization steps (99.3%). Notably, FOMAML's meta-initialization alone matches the performance of 10 adaptation steps, indicating the primary benefit comes from meta-training rather than per-task adaptation. (b) Optimization efficiency to reach 98.9% fidelity: FOMAML achieves this with zero per-task optimization steps (meta-initialization only), while GRAPE from scratch fails to reach this threshold even after 150 iterations (achieving only 97.5%). Per-task GRAPE requires warm-starting and 200 steps to surpass FOMAML. This highlights that the meta-learned initialization provides a superior starting point that GRAPE cannot match through optimization alone, directly addressing the calibration bottleneck where rapid deployment without per-device optimization is valuable.

## A.6: Tunable Coupler CZ Gate

To demonstrate that our scaling laws generalize beyond noise-rate variation to Hamiltonian parameter variation, we validate on two-qubit CZ gates with tunable ZZ coupling.

**Physical motivation.** Modern superconducting quantum processors employ tunable couplers to enable high-fidelity two-qubit gates while suppressing always-on ZZ interactions during idle periods (Yan et al., 2018). In these architectures, the effective coupling strength $J$ between qubits can be dynamically adjusted via flux-tunable coupler elements. However, the achievable coupling range and residual ZZ interactions vary across qubit pairs due to:

- **Fabrication variation:** Junction asymmetries and lithographic tolerances cause $\pm 10$–$20\%$ variation in coupling strengths across nominally identical qubit pairs on the same chip.

- **Frequency crowding:** In multi-qubit processors, qubit frequencies must be staggered to avoid collisions, leading to different detunings and effective couplings for each pair.

**System Hamiltonian.** We extend the two-qubit Hamiltonian (Eq. A.48) with a controllable coupling term:

$$H(t) = J\,\sigma_1^z \otimes \sigma_2^z + u_{ZZ}(t)\,\sigma_1^z \otimes \sigma_2^z + \sum_{j=1}^{2} \left[ u_{x,j}(t)\sigma_j^x + u_{y,j}(t)\sigma_j^y + u_{z,j}(t)\sigma_j^z \right], \tag{A.1}$$

which simplifies to:

$$H(t) = \left(J + u_{ZZ}(t)\right)\sigma_1^z \otimes \sigma_2^z + \sum_{j=1}^{2} \left[ u_{x,j}(t)\sigma_j^x + u_{y,j}(t)\sigma_j^y + u_{z,j}(t)\sigma_j^z \right]. \tag{A.2}$$

Here $J$ is the static (task-dependent) coupling strength and $u_{ZZ}(t)$ is a time-dependent control field that modulates the effective coupling. This yields seven control channels: $\{u_{x,1}, u_{y,1}, u_{z,1}, u_{x,2}, u_{y,2}, u_{z,2}, u_{ZZ}\}$.

**Task distribution.** Tasks are defined by the static coupling strength $J \in \{1.0, 3.0, 6.0, 9.0\}$ (arbitrary units), sampled uniformly. This range reflects realistic device heterogeneity where coupling strengths vary by factors of 2–3$\times$ across qubit pairs. Unlike the noise-variation experiments, dissipation rates are held fixed at $\Gamma_{\text{deph}} = 0.005$ and $\Gamma_{\text{relax}} = 0.0025$ for both qubits.

**Control strategy.** The key insight is that $u_{ZZ}(t)$ allows the controller to *compensate* for unknown static coupling $J$. For weak native coupling ($J = 1.0$), the controller must supply additional ZZ interaction via $u_{ZZ}(t) > 0$ to accumulate sufficient entangling phase. For strong native coupling ($J = 9.0$), the controller may need to partially cancel the interaction or modulate timing to avoid over-rotation. This creates a natural test of control separation (Lemma 4.7): different values of $J$ require qualitatively different control strategies.

**Results.** Figure A.6 shows that:

1. The adaptation gap follows exponential saturation from $\mathcal{F}_0 = 0.41$ to $\mathcal{F} \approx 0.99$ after $K = 30$ steps.

2. Adapted $u_{ZZ}$ pulses exhibit task-specific structure: weak coupling requires large corrective amplitudes while strong coupling requires dynamic modulation.

3. The full seven-channel control waveforms (panels c to f) demonstrate control separation: each value of $J$ yields a distinct optimal pulse sequence.

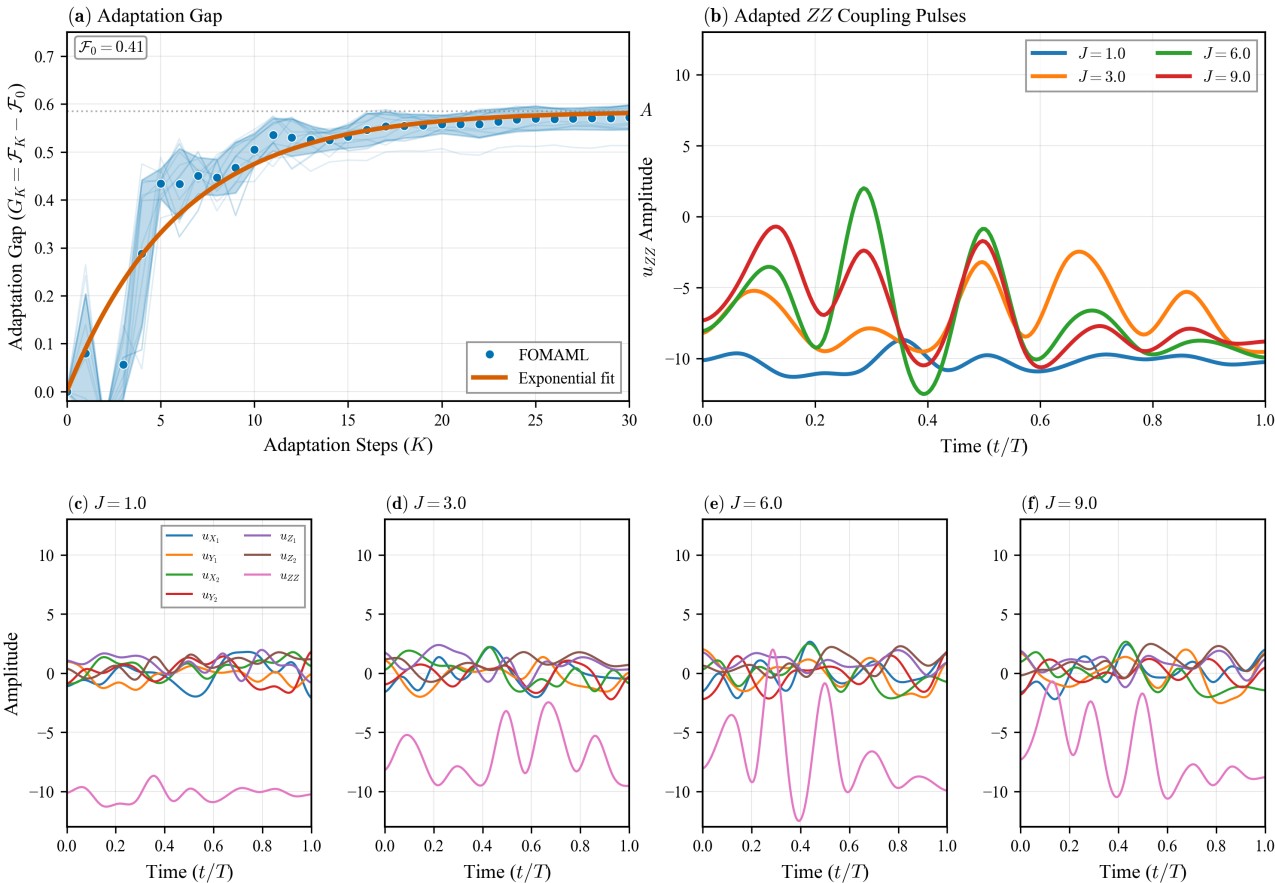

*Figure A.6.* **Scaling law validation on Hamiltonian parameter variation.** Two-qubit CZ gate with tunable coupling strength $J \in \{1.0, 3.0, 6.0, 9.0\}$ as the task parameter. (a) Adaptation gap follows exponential saturation from $\mathcal{F}_0 = 0.41$ to $\mathcal{F} \approx 0.99$. (b) Adapted $u_{ZZ}$ pulses show task-specific structure: weak coupling ($J = 1.0$) requires maximal correction while strong coupling requires dynamic modulation. (c–f) Full seven-channel control waveforms for each $J$ value, demonstrating control separation across Hamiltonian parameter variation.

These results confirm that the scaling law $\sigma_\tau^2(1 - e^{-\beta K})$ holds for Hamiltonian parameter variation, not just noise-rate variation.

## A.7: PL Condition Breadth Across Tasks

The PL constant $\mu(\xi)$ governs the rate $\beta = \eta\mu_{\min}$ in Theorem 4.8, so understanding how $\mu$ varies across the task distribution and under distribution shift is essential for assessing where the scaling law applies.

We characterize the PL condition's breadth across the task distribution by computing $\mu(\xi)$ for each task from the locally linear regime of $\frac{1}{2}\|\nabla L\|^2$ vs. $L - L^*$ near optimum.[1]

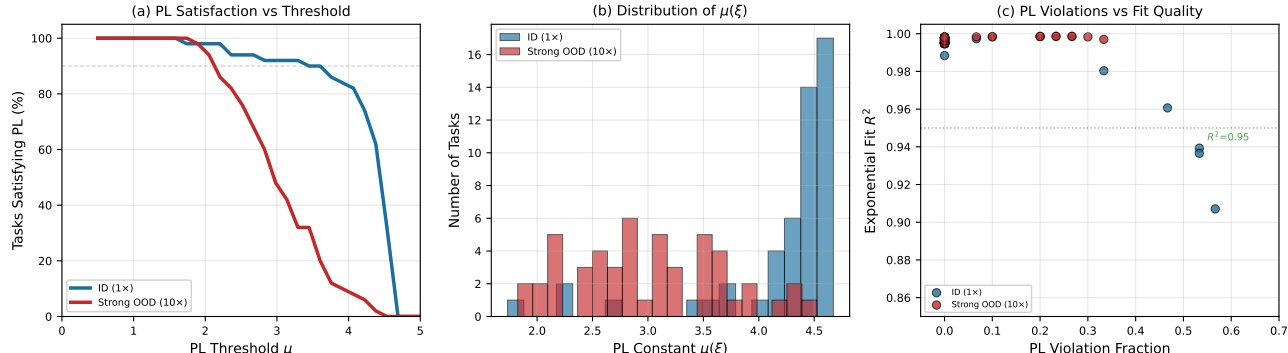

*Figure A.7.* **PL breadth across the task distribution.** (a) Fraction of tasks satisfying PL above threshold $\mu$, for ID (1×) and strong OOD (10×). (b) Distribution of $\mu(\xi)$: ID concentrates near $\mu \approx 4.4$; strong OOD shifts toward smaller $\mu$. (c) Exponential-fit $R^2$ vs. PL violation fraction: even at $60\%$ violation, the median fit retains $R^2 > 0.95$.

PL satisfaction is universal at moderate thresholds for ID and remains high under strong OOD (panel a). Median $\mu(\xi)$ shifts from $\approx 4.4$ to $\approx 2.9$ under strong OOD (panel b). Most importantly, the scaling-law fit $R^2 > 0.95$ across all observed PL violation rates (panel c).

**Takeaways.** *(i) Universal satisfaction at moderate threshold.* Panel (a) shows that PL holds for essentially all tasks at thresholds $\mu \lesssim 1.5$ in both ID and strong OOD regimes, validating Assumption 4.3 as a working operating-regime condition rather than an idealization. *(ii) Graceful curvature degradation under shift.* Panel (b) shows the $\mu(\xi)$ distribution shifts from a tight ID concentration near $\mu \approx 4.4$ toward a broader strong-OOD distribution with median $\mu \approx 2.9$ and a heavier left tail. The shift is gradual, not catastrophic: optimization geometry weakens under distribution shift but does not break. *(iii) Robustness of the scaling-law form.* Panel (c) plots the exponential-fit $R^2$ on $G_K$ against the PL violation fraction (computed by sweeping the threshold $\mu^*$ and recording, for each threshold, the fraction of tasks falling below it). Across both regimes, $R^2$ remains above $0.9$ even when more than half the tasks violate the strict PL threshold, confirming that the scaling-law form persists outside the strict PL basin. The slightly lower $R^2$ values for ID points relative to strong OOD reflect a signal-to-noise effect—absolute gap magnitudes in the ID regime are small ($A_\infty \sim 10^{-3}$), so fixed-amplitude residuals appear more prominently in the relative fit metric—rather than any degradation of the scaling law in the well-behaved regime. The robustness in panel (c) is consistent with the polynomial-decay generalization derived under the weaker Kurdyka–Łojasiewicz condition.

**Practical diagnostic.** For new systems, PL validity is assessed from the first 3–5 probe steps used by Algorithm 1: $R^2 < 0.9$ on the exponential fit signals departure from the local PL basin and unreliable budget predictions.

---

[1] Aggregate $\mu$ values use a fixed loss-gap window normalized per task and so differ in scale from the example $\mu \approx 0.03$ in Figure 3(a), which is fit over a wider window.

## A.8: Few-Shot Decision Protocol

The few-shot protocol (Algorithm 1) hinges on whether the exponential model $G_K \approx A_\infty(1 - e^{-\beta K})$ can be reliably fit from very few probe steps. We characterize this in two ways: (i) visually, by overlaying few-shot fits on the full-data curve as the probe budget $N$ grows; and (ii) numerically, by reporting the relative error of $\hat{K}_{0.95}$ across out-of-distribution severity. Figure A.8 and Table A.4 report both views.

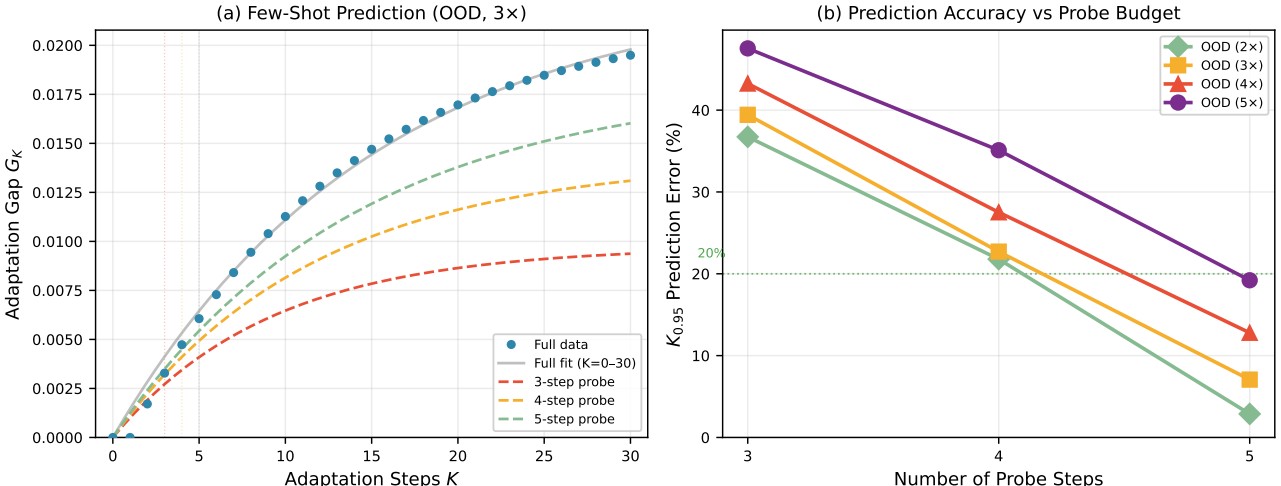

*Figure A.8.* **Few-shot pre-adaptation prediction.** (a) On an OOD ($3\times$) condition, $N$-step probe fits (dashed) converge to the full-data fit (gray) as $N$ grows. (b) Relative error of $\hat{K}_{0.95}$ across OOD severity; error decays roughly $5\times$ from $N=3$ to $N=5$ and stays below the $20\%$ practitioner threshold for $N=5$ at moderate OOD.

*Table A.4.* Few-shot budget prediction. $K_{0.95}$ is the true budget; subsequent columns report the relative error of $\hat{K}_{0.95}$.

| Regime | $K_{0.95}$ | $N=3$ | $N=4$ | $N=5$ |
|---|---|---|---|---|
| OOD ($2\times$) | 43.3 | 36.7% | 21.8% | 2.9% |
| OOD ($3\times$) | 45.4 | 39.4% | 22.7% | 7.1% |
| OOD ($4\times$) | 48.4 | 43.2% | 27.5% | 12.8% |
| OOD ($5\times$) | 52.4 | 47.5% | 35.1% | 19.2% |

**Practitioner threshold.** Panel (b) marks $20\%$ as a practitioner threshold: errors below this level let the operator round to a standard budget choice without changing the outcome. With $N = 5$ probes, all four OOD regimes meet this threshold. When the protocol returns an estimate near a budget boundary, the simplest fallback is to run two or three additional probe steps and refit, which empirically tightens the estimate further at the cost of a few percent of total adaptation budget.

Accuracy improves smoothly with $N$ and degrades gradually with OOD severity rather than collapsing. Independently of probes, $\hat{A}_\infty$ can be estimated *a priori* from $\sigma_\tau^2$ (Figure 4b, $R^2 = 0.94$) using $T_1/T_2$ characterization, providing a zero-shot decision signal.

## A.9: Disentangling Conditioning from Adaptation

Because the policy $\pi_\theta$ takes $\xi$ as context input (Section 5.1), the adaptation gains reported in Sections 5.2–5.3 could in principle stem from contextual conditioning rather than gradient-based adaptation. We isolate the two mechanisms by an ablation that crosses $\xi$-access against gradient adaptation, with results in Figure A.9 and Table A.5. We cross $\xi$-access (none, full, 30% noisy) with adaptation (none, $K{=}30$). All conditions share architecture and optimizer; "Blind" rows replace task features with zero, and "Noisy $\xi$" uses $\xi(1 + \epsilon)$ with $\epsilon \sim \mathcal{N}(0, 0.3^2)$, modeling realistic $T_1/T_2$ uncertainty.

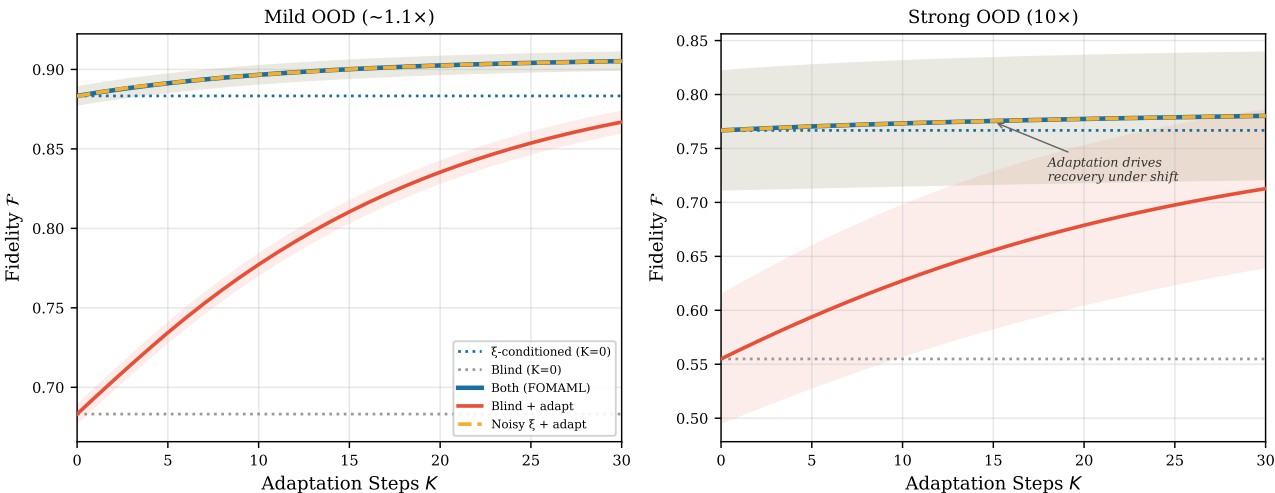

*Figure A.9.* **Disentangling conditioning from adaptation.** Fidelity trajectories under (left) mild OOD and (right) strong OOD ($10\times$). Dotted: pre-adaptation baselines. Red: blind + adapt. Blue: full FOMAML. Dashed yellow (noisy $\xi$ + adapt) overlaps the full method. Adaptation alone closes the bulk of the gap to the full method under both shifts.

*Table A.5.* Disentangling conditioning from adaptation. Adaptation alone (no $\xi$) contributes $+0.16$–$0.18$ over the blind baseline; noisy $\xi$ matches the full method.

| Method | $\xi$ | Adapt | Mild OOD | Strong OOD |
|---|---|---|---|---|
| Blind | – | – | 0.683 | 0.555 |
| $\xi$-only | full | – | 0.883 | 0.767 |
| Adapt-only | – | $K{=}30$ | 0.867 | 0.713 |
| Noisy $\xi$ + adapt | noisy | $K{=}30$ | 0.905 | 0.780 |
| Both (FOMAML) | full | $K{=}30$ | **0.905** | **0.780** |

**Three observations.** *(i) Adaptation alone closes most of the gap.* Starting from a blind initialization (no $\xi$ access), $K{=}30$ gradient steps deliver $+0.184$ (mild OOD) and $+0.158$ (strong OOD) over the blind baseline. This is the central finding: adaptation, by itself, recovers the bulk of the achievable fidelity gain without any task-descriptor access. *(ii) Conditioning is complementary, not essential.* $\xi$-only (no gradient steps) raises the pre-adaptation fidelity by $+0.200$ (mild)/ $+0.212$ (strong), but never reaches the full-method fidelity under either shift. Conditioning provides a better starting point; adaptation closes the remaining distance. *(iii) Robustness to imperfect $\xi$.* Replacing the true task descriptor with $\xi(1 + \epsilon)$, $\epsilon \sim \mathcal{N}(0, 0.3^2)$, matches the full method to within reporting precision (dashed yellow overlapping blue in Figure A.9), supporting deployment workflows where $T_1/T_2$ characterization is imperfect.

**Implications.** The central claim of the paper (that adaptation gains scale with task variance via Theorem 4.8) is therefore not an artifact of contextual conditioning. Even stripped of $\xi$ entirely, the gradient steps themselves deliver gap-closing recovery whose magnitude tracks the predicted scaling, ruling out the alternative interpretation that the few-shot protocol merely exploits well-engineered task embeddings.

## Appendix B: Algorithms

We provide pseudocode for the meta-learning procedure (Algorithm 2) and the inner-loop adaptation subroutine (Algorithm 3), complementing the few-shot decision protocol (Algorithm 1) in the main text. Gradients $\partial\rho(T)/\partial\theta$ are computed by backpropagation through a 4th-order Runge-Kutta integrator; we use FOMAML (Nichol et al., 2018) for the outer loop.

---

**Algorithm 2** Meta-Learning for Quantum Control

---

**Require:** Task distribution $P(\xi)$ and Lindbladian generators $f_\xi = \mathcal{L}_\xi$, number of meta-iterations $M$, inner-loop steps $K$, learning rates $\eta_{\text{in}}, \eta_{\text{out}}$

1: Initialize meta-parameters (control waveform initialization) $\theta_0$
2: **for** each meta-iteration $m = 1, \ldots, M$ **do**
3:    Sample a batch of $N$ tasks $\{\xi_i\}_{i=1}^N \sim P(\xi)$
4:    **for** each task $\xi_i$ in parallel **do**
5:       Initialize task-specific parameters $\theta_i^{(0)} \leftarrow \theta_{m-1}$
6:       **for** inner step $k = 1, \ldots, K$ **do**
7:          Propagate Lindblad dynamics under control $u(t; \theta_i^{(k-1)})$:
8:             $\dot\rho(t) = \mathcal{L}_\xi(\rho(t); u(t; \theta_i^{(k-1)}))$
9:          Compute fidelity loss at final time:
10:             $L(\theta_i^{(k-1)}; \xi_i) = 1 - F(\rho(T), \rho_{\text{target}})$
11:          Compute gradient via automatic differentiation:
12:             $\nabla_{\theta_i} L = \frac{\partial L}{\partial \rho(T)} \cdot \frac{\partial \rho(T)}{\partial \theta_i}$
13:          Update control parameters (inner adaptation):
14:             $\theta_i^{(k)} \leftarrow \theta_i^{(k-1)} - \eta_{\text{in}} \nabla_{\theta_i} L(\theta_i^{(k-1)}; \xi_i)$
15:       **end for**
16:       Compute post-adaptation loss on the same task: $\tilde{L}(\theta_i^{(K)}; \xi_i)$
17:    **end for**
18:    Compute meta-gradient using FOMAML (treating $\theta_i^{(K)}$ as independent of $\theta_{m-1}$):
19:       $g_{\text{meta}} = \frac{1}{N} \sum_{i=1}^N \nabla_\theta \tilde{L}(\theta; \xi_i)\big|_{\theta=\theta_i^{(K)}}$
20:    Update meta-parameters (outer update):
21:       $\theta_m \leftarrow \theta_{m-1} - \eta_{\text{out}} g_{\text{meta}}$
22: **end for**
    Return: Meta-initialized parameters $\theta_m$ for rapid task-specific adaptation)

---

**Algorithm 3** Inner-loop task adaptation via differentiable quantum control

---

**Require:** Task $\xi$ and Lindbladian generator $f_\xi = \mathcal{L}_\xi$), initial control parameters $\theta^{(0)}$ (from meta-init), number of inner steps $K$, step size $\eta_{\text{in}}$
**Ensure:** Adapted control parameters $\theta^{(K)}$ specialized to task $\xi$

1: **for** $k = 1, \ldots, K$ **do**
2:    **Forward propagate dynamics:**
3:       Initialize density matrix $\rho(0)$
4:       Evolve under current control pulses $u(t; \theta^{(k-1)})$:
5:          $\frac{d\rho(t)}{dt} = \mathcal{L}_\xi\left(\rho(t); u(t; \theta^{(k-1)})\right), \quad t \in [0, T]$
6:       Obtain final state $\rho(T)$
7:    **Compute task loss:**
8:       $L(\theta^{(k-1)}; \xi) = 1 - F(\rho(T), \rho_{\text{target}})$
9:    **Compute gradient via automatic differentiation:**
10:       $\nabla_\theta L(\theta^{(k-1)}; \xi) = \frac{\partial L}{\partial \rho(T)} \cdot \frac{\partial \rho(T)}{\partial \theta^{(k-1)}}$
11:       where $\frac{\partial \rho(T)}{\partial \theta}$ is computed by backpropagating through the
12:       differentiable ODE integrator (with automatic differentiation)
13:    **Gradient descent update (inner-loop step):**
14:       $\theta^{(k)} \leftarrow \theta^{(k-1)} - \eta_{\text{in}} \nabla_\theta L(\theta^{(k-1)}; \xi)$
15: **end for**
    Return adapted parameters $\theta^{(K)}$

---

## Appendix C: Extended Theoretical Analysis

This appendix provides proofs of our main theoretical results.

### C.1: Proofs of Main Lemmas

PROOF OF LEMMA 4.4

*Proof Sketch.* Under the controllability and unconstrained-control assumptions, the end-point map from controls to the final unitary propagator is locally surjective onto $\mathrm{SU}(d)$.

The kinematic analysis of Russell *et al.* (Russell et al., 2017) shows that for such systems all critical points of the fidelity objective correspond to either global maxima or saddle points, this implies the absence of generically suboptimal local maxima.

Russell, Rabitz, and Wu (Russell et al., 2017) strengthen this conclusion using parametric transversality, proving that trap-free landscapes hold for almost all Hamiltonians and that singular controls do not generically create traps.

**Remark (Open-system caveat).** For Lindblad dynamics the reachable set and end-point map geometry change qualitatively, and a general trap-free theorem is not known. Structural conditions such as those in Baggio and Viola (Baggio et al., 2021) can guarantee attractive behavior for specific objectives (a basin of attraction), but this itself does not imply global trap-free landscapes for a generic open quantum system. In this work we rely on the rigorous closed-system result together with continuity arguments and empirical validation for tasks with weak and smoothly varying dissipation. □

PROOF OF LEMMA 4.5

*Proof Sketch.* The Lindblad generator acting on density matrix $\rho$ is:

$$\mathcal{L}_\xi[\rho] = -i[H_\xi, \rho] + \sum_j \Gamma_j(\xi)\tilde{\mathcal{D}}[\sigma_j]\rho \tag{A.3}$$

where $\tilde{\mathcal{D}}[\sigma_j]\rho = \sigma_j\rho\sigma_j^\dagger - \frac{1}{2}\{\sigma_j^\dagger\sigma_j, \rho\}$ is the *normalized* dissipator (independent of $\xi$). See Appendix D for notation and definitions.

**Case 1: Dissipation rate variation**

When the Hamiltonian is task-independent ($H_\xi = H_0$) and $\xi = (\Gamma_{\mathrm{deph}}, \Gamma_{\mathrm{relax}})$:

$$\mathcal{L}_\xi[\rho] - \mathcal{L}_{\xi'}[\rho] = \sum_j (\Gamma_j(\xi) - \Gamma_j(\xi'))\tilde{\mathcal{D}}[\sigma_j]\rho \tag{A.4}$$

Since the normalized dissipator $\tilde{\mathcal{D}}[\sigma_j]$ is a bounded linear superoperator with $\|\tilde{\mathcal{D}}[\sigma_j]\| \leq C_j$ for some constant $C_j$ depending on the jump operator structure, we have:

$$\|\mathcal{L}_\xi - \mathcal{L}_{\xi'}\| \leq \sum_j |\Gamma_j - \Gamma_j'| \cdot \|\tilde{\mathcal{D}}[\sigma_j]\| \leq C_L\|\xi - \xi'\| \tag{A.5}$$

**Case 2: Hamiltonian parameter variation**

When $\xi$ parameterizes Hamiltonian coefficients (e.g., coupling strength $J$ in $H_\xi = J\sigma_z \otimes \sigma_z + H_{\mathrm{ctrl}}$), the generator difference includes a commutator term:

$$\mathcal{L}_\xi[\rho] - \mathcal{L}_{\xi'}[\rho] = -i[(H_\xi - H_{\xi'}), \rho] + \text{(dissipator terms)} \tag{A.6}$$

The commutator satisfies $\|[A, \rho]\| \leq 2\|A\|\|\rho\|$, giving:

$$\| - i[(H_\xi - H_{\xi'}), \rho]\| \leq 2\|H_\xi - H_{\xi'}\| \leq C_H|\xi - \xi'| \tag{A.7}$$

where $C_H$ depends on the Hamiltonian structure (e.g., $C_H = 2\|\sigma_z \otimes \sigma_z\| = 2$ for coupling variation).

Combining both contributions, we obtain in the generic notation of the main text (setting $f_\xi = \mathcal{L}_\xi$):

$$\|f_\xi - f_{\xi'}\| \leq C_L \|\xi - \xi'\| \tag{A.8}$$

where $C_L > 0$ aggregates constants from dissipator and/or Hamiltonian contributions depending on the task parameterization. □

PROOF OF LEMMA 4.7

*Proof.* Let $\theta^*(\xi)$ denote the optimal control parameters for task $\xi \in \Xi$. Fix a reference task $\xi^* \in \Xi$ with optimum $\theta^* = \theta^*(\xi^*)$. Here we assume the following: (1) $L_\xi(\theta)$ is twice continuously differentiable in $(\theta, \xi)$, (2) the Hessian $H = \nabla^2_{\theta\theta} L_{\xi^*}(\theta^*)$ satisfies $H \succeq \mu I$ for some $\mu > 0$, and (3) the mixed partial $M = \nabla^2_{\xi\theta} L_{\xi^*}(\theta^*)$ satisfies $\sigma_{\min}(M) > 0$ (full column rank), where $\sigma_{\min}(M)$ is the smallest singular value of $M$.

By the implicit function theorem applied to the optimality condition $\nabla_\theta L_\xi(\theta^*(\xi)) = 0$, the mapping $\xi \mapsto \theta^*(\xi)$ is differentiable with the Jacobian:

$$\left.\frac{\partial\theta^*}{\partial\xi}\right|_{\xi^*} = -H^{-1}M. \tag{A.9}$$

A Taylor expansion around $\xi^*$ leads to the relation:

$$\theta^*(\xi) - \theta^*(\xi^*) = -H^{-1}M(\xi - \xi^*) + O(\|\xi - \xi^*\|^2). \tag{A.10}$$

Since $\dim(\theta) \geq \dim(\xi)$, the matrix $J = -H^{-1}M$ has dimensions $\dim(\theta) \times \dim(\xi)$ and can have full column rank. The condition $\sigma_{\min}(M) > 0$ ensures $M$ has full column rank, and since $H \succeq \mu I$ is invertible, $J = -H^{-1}M$ also has full column rank.

The result follows for $\|\delta\xi\|$ sufficiently small that the linear term dominates. □

*Verification for quantum control*: For quantum gate synthesis with task parameters $\xi = (\Gamma_{\text{deph}}, \Gamma_{\text{relax}})$ ($\dim(\xi) = 2$) and control parameters $\theta$ representing pulse amplitudes ($\dim(\theta) \geq 2$ in all experiments), the dimension condition is satisfied. The task sensitivity condition $\sigma_{\min}(M) > 0$ holds generically under controllability. Physically, distinct noise environments require distinct optimal pulses, and this dependence is first-order except at measure-zero degenerate points. We verify $C_{\text{sep}} > 0$ numerically in Figure 3(c).

## C.2: Proof of Main Theorem (Theorem 4.8)

We prove the adaptation gap scaling law $G_K \geq A_\infty(1 - e^{-\beta K})$.

SETUP

The adaptation gap is defined as:

$$G_K = \mathbb{E}_{\xi \sim P}[L_\xi(\theta_0) - L_\xi(\theta_K(\xi))],$$

where $\theta_0$ is the meta-initialization and $\theta_K(\xi)$ is the adapted policy after $K$ gradient steps on task $\xi$.

PART A: PL CONVERGENCE.

The functional form of the gap depends on the loss surface obeying the PL condition locally near the task optima. We state the proof of this fact in terms of loss surfaces satisfying the weaker Kurdyka-Łojasiewicz (KL) condition of which the PL condition is a special case. Formally we assume here that for some $\alpha \geq 1/2$ the loss surface satisfies

$$\|\nabla_\theta L_\xi(\theta)\|^2 \geq \tilde{c}(L_\xi(\theta) - L_\xi^*)^{2\alpha} \tag{A.11}$$

in a neighborhood containing the optimum $\theta_\xi^*$ and $\theta_0$. Note that if $\alpha = 1/2$ this is equivalent to the PL condition.

Gradient descent yields the standard recurrence relation for L-smooth functions,

$$L_\xi(\theta_{k+1}) \leq L_\xi(\theta_k) - \eta\|\nabla L_\xi(\theta_k)\|^2 + \frac{L\eta^2}{2}\|\nabla L_\xi(\theta_k)\|^2. \tag{A.12}$$

By the KL condition and assuming that $\eta \leq 1/L$ (where L is the Lipschitz constant of $\nabla L_\xi$),

$$L_\xi(\theta_{k+1}) - L_\xi^* \leq L_\xi(\theta_k) - L_\xi^* - \frac{\eta}{2}\tilde{c}(L_\xi(\theta_k) - L_\xi^*)^{2\alpha}. \tag{A.13}$$

Treating $k$ as continuous and taking the continuous limit gives the associated ODE

$$\Delta_{k+1} \leq \Delta_k - \gamma\Delta_k^{2\alpha}, \tag{A.14}$$

where $\gamma = \eta\tilde{c}/2$.

$$\frac{d\Delta}{dt} = -\gamma\Delta(t)^{2\alpha}. \tag{A.15}$$

Solving this ordinary differential equation gives

$$\Delta(t) = \begin{cases} [\Delta_0^{1-2\alpha} - (1-2\alpha)\gamma t]^{\frac{-1}{2\alpha-1}} & \alpha > 1/2 \\ \exp(-\gamma t) & \alpha = 1/2 \end{cases}. \tag{A.16}$$

For $\alpha = 1/2$ we have the special case of the PL condition and we have the inequality

$$L_\xi(\theta_K) - L_\xi^* \leq \exp(-\beta K)[L_\xi(\theta_0) - L_\xi^*], \tag{A.17}$$

where $\beta = \eta\mu$.

Though not used in the main body of this paper, one can also derive a bound on the rate of gradient descent convergence under the more general KL condition and arrive at

$$L_\xi(\theta_k) - L_\xi^* \leq \left((L_\xi(\theta_0) - L_\xi^*)^{1-2\alpha} - (1-2\alpha)\gamma k\right)^{\frac{-1}{2\alpha-1}}. \tag{A.18}$$

PART B: GAP DECOMPOSITION.

We decompose the adaptation gap into the initial suboptimality minus the residual suboptimality after $K$ steps, then apply the PL convergence bound to obtain the exponential saturation form.

$$L_\xi(\theta_0) - L_\xi(\theta_K) = [L_\xi(\theta_0) - L_\xi^*] - [L_\xi(\theta_K) - L_\xi^*] \tag{A.19}$$

$$\geq [L_\xi(\theta_0) - L_\xi^*](1 - e^{-\beta K}). \tag{A.20}$$

PART C: EXPECTATION.

Define $\varepsilon_{\text{init}} := \mathbb{E}_\xi[L_\xi(\theta_0) - L_\xi^*]$. Taking expectation values:

$$G_K \geq \varepsilon_{\text{init}}(1 - e^{-\beta K}). \tag{A.21}$$

For MAML-optimized $\theta_0$, this bound is approximately tight.

PART D: CONNECTING TO TASK VARIANCE.

We establish $A_\infty \propto \sigma_\tau^2$ in two steps: (i) showing that MAML and average-task solutions coincide under appropriate conditions, and (ii) relating the initial suboptimality at this solution to task variance.

**Step 1: MAML and average-task solutions coincide under symmetry.** Consider quadratic task losses of the form:

$$L_\xi(\theta) = \frac{1}{2}(\theta - \theta^*(\xi))^\top H(\theta - \theta^*(\xi)) \tag{A.22}$$

with Hessian $H \succ 0$ and linear optima map $\theta^*(\xi) = A\xi + b$.

The task-independent Hessian is a local approximation: in principle $H = H(\xi)$, but for tasks within a bounded neighborhood of the mean $\bar{\xi}$, we have $H(\xi) = H(\bar{\xi}) + O(\|\xi - \bar{\xi}\|)$. When task variance $\sigma_\tau^2$ is moderate, this first-order correction contributes

$O(\sigma_\tau^3)$ to the final bound, which is dominated by the leading $O(\sigma_\tau^2)$ term. This approximation is consistent with our empirical observation that the PL constant $\mu$ varies weakly across tasks within the training distribution (Figure 3(a) shows similar slopes for different tasks in the adaptation regime, with $\mu \approx 0.03$ for both).

Previous work (Arnold et al., 2021) has proved for 1D linear regression that the MAML solution equals the average-task solution under symmetric task distributions. We extend their argument to our setting.[2]

The MAML objective with one inner-loop step is:

$$\mathcal{L}^{\text{MAML}}(\theta) = \mathbb{E}_\xi \left[ L_\xi(\theta - \eta \nabla_\theta L_\xi(\theta)) \right] \tag{A.23}$$

The gradient of the quadratic loss is $\nabla_\theta L_\xi(\theta) = H(\theta - \theta^*(\xi))$, so the inner-loop update gives:

$$\theta' = \theta - \eta H(\theta - \theta^*(\xi)) = (I - \eta H)\theta + \eta H \theta^*(\xi) \tag{A.24}$$

The post-adaptation residual is after $K = 1$ adaptation steps is:

$$\begin{aligned}
\theta' - \theta^*(\xi) &= (I - \eta H)\theta + \eta H \theta^*(\xi) - \theta^*(\xi) \\
&= (I - \eta H)\theta - (I - \eta H)\theta^*(\xi) \\
&= (I - \eta H)(\theta - \theta^*(\xi))
\end{aligned} \tag{A.25}$$

Substituting into the loss:

$$\begin{aligned}
L_\xi(\theta') &= \frac{1}{2}(\theta' - \theta^*(\xi))^\top H(\theta' - \theta^*(\xi)) \\
&= \frac{1}{2}\left[(I - \eta H)(\theta - \theta^*(\xi))\right]^\top H \left[(I - \eta H)(\theta - \theta^*(\xi))\right] \\
&= \frac{1}{2}(\theta - \theta^*(\xi))^\top M(\theta - \theta^*(\xi))
\end{aligned} \tag{A.26}$$

where $M = (I - \eta H)^\top H(I - \eta H)$.

To compute $\mathcal{L}^{\text{MAML}}(\theta) = \mathbb{E}_\xi[L_\xi(\theta')]$, let $\bar{\theta}^* = \mathbb{E}_\xi[\theta^*(\xi)]$ and decompose:

$$\theta - \theta^*(\xi) = (\theta - \bar{\theta}^*) + (\bar{\theta}^* - \theta^*(\xi)) \tag{A.27}$$

Expanding the quadratic form:

$$\begin{aligned}
(\theta - \theta^*(\xi))^\top M(\theta - \theta^*(\xi)) = {}&(\theta - \bar{\theta}^*)^\top M(\theta - \bar{\theta}^*) \\
&+ 2(\theta - \bar{\theta}^*)^\top M(\bar{\theta}^* - \theta^*(\xi)) \\
&+ (\bar{\theta}^* - \theta^*(\xi))^\top M(\bar{\theta}^* - \theta^*(\xi))
\end{aligned} \tag{A.28}$$

Taking expectations term by term:

*Term 1:* $(\theta - \bar{\theta}^*)^\top M(\theta - \bar{\theta}^*)$ contains no dependence on $\xi$, so:

$$\mathbb{E}_\xi \left[ (\theta - \bar{\theta}^*)^\top M(\theta - \bar{\theta}^*) \right] = (\theta - \bar{\theta}^*)^\top M(\theta - \bar{\theta}^*) \tag{A.29}$$

*Term 2:* Pulling out constants and using $\bar{\theta}^* = \mathbb{E}_\xi[\theta^*(\xi)]$:

$$\mathbb{E}_\xi \left[ 2(\theta - \bar{\theta}^*)^\top M(\bar{\theta}^* - \theta^*(\xi)) \right] = 2(\theta - \bar{\theta}^*)^\top M \cdot \mathbb{E}_\xi \left[ \bar{\theta}^* - \theta^*(\xi) \right] = 0 \tag{A.30}$$

*Term 3:* Using the trace identity $a^\top B a = \text{Tr}(B a a^\top)$ and the definition of covariance:

$$\begin{aligned}
\mathbb{E}_\xi \left[ (\bar{\theta}^* - \theta^*(\xi))^\top M(\bar{\theta}^* - \theta^*(\xi)) \right] &= \text{tr}\left( M \cdot \mathbb{E}_\xi \left[ (\theta^*(\xi) - \bar{\theta}^*)(\theta^*(\xi) - \bar{\theta}^*)^\top \right] \right) \\
&= \text{tr}(M \Sigma_{\theta^*})
\end{aligned} \tag{A.31}$$

---

[2]The equivalence is established for $K = 1$; for $K > 1$ with small $\eta$, the MAML objective remains approximately minimized at the average-task solution, as higher-order corrections scale with $O(\eta^2)$.

Combining and multiplying by $\frac{1}{2}$:

$$\mathcal{L}^{\mathrm{MAML}}(\theta) = \frac{1}{2}(\theta - \bar{\theta}^*)^\top M(\theta - \bar{\theta}^*) + \frac{1}{2}\mathrm{tr}(M\Sigma_{\theta^*}) \tag{A.32}$$

The second term is constant in $\theta$, so:

$$\theta_0^{\mathrm{MAML}} = \bar{\theta}^* = \mathbb{E}_\xi[\theta^*(\xi)] \tag{A.33}$$

An identical calculation for the average-task objective $\mathbb{E}_\xi[L_\xi(\theta)]$ (setting $\eta = 0$, hence $M = H$) yields $\theta_{\mathrm{avg}} = \mathbb{E}_\xi[\theta^*(\xi)]$. Thus $\theta_0^{\mathrm{MAML}} = \theta_{\mathrm{avg}}$ under the quadratic-linear structure, independent of learning rate $\eta$ (provided $\eta < 1/\|H\|$ for stability).

**Step 2: Initial suboptimality scales with task variance.** At $\theta_0 = \mathbb{E}_\xi[\theta^*(\xi)]$, the expected initial suboptimality is:

$$A_\infty = \mathbb{E}_\xi[L_\xi(\theta_0) - L_\xi^*] = \frac{1}{2}\mathrm{tr}(H\Sigma_{\theta^*}) \tag{A.34}$$

By the linear optima map $\theta^*(\xi) = A\xi + b$ (validated empirically in 3(c)):

$$\Sigma_{\theta^*} = A \cdot \mathrm{Cov}(\xi) \cdot A^\top \tag{A.35}$$

Substituting:

$$A_\infty = \frac{1}{2}\mathrm{tr}(A^\top HA \cdot \mathrm{Cov}(\xi)) \tag{A.36}$$

For isotropic task variance $\mathrm{Cov}(\xi) = \sigma_\tau^2 I / \dim(\xi)$:

$$A_\infty = \frac{\sigma_\tau^2}{2 \cdot \dim(\xi)}\mathrm{tr}(A^\top HA) = c \cdot \sigma_\tau^2 \tag{A.37}$$

where $c = \mathrm{tr}(A^\top HA)/(2 \cdot \dim(\xi))$ depends on loss geometry ($H$) and control separation ($A$).

**Assumptions.** This derivation relies on: (i) quadratic loss approximation near optima, (ii) approximately task-independent Hessian, and (iii) linear dependence of optima on task parameters.

Combining Parts A–D:

$$\boxed{G_K \geq c\,\sigma_\tau^2(1 - e^{-\beta K}), \quad \beta = \eta\mu_{\min}.} \tag{A.38}$$

Figure 4(b) validates $A_\infty \propto \sigma_\tau^2$ with $R^2 = 0.94$ for FOMAML-trained initializations.

Note that a weaker result can be arrived at under the KL condition by replacing the bound in (A.19) with the bound derived from (A.17). In this case, the remainder of the argument remains unchanged and you arrive at a bound on the gap of the form

$$\boxed{G_k \geq c\sigma_\tau^2(1 - \mathcal{O}((\gamma k)^{\frac{-1}{2\alpha-1}})), \quad \gamma = \eta\tilde{c}/2} \tag{A.39}$$

$\square$

### C.3: Empirical Verification

Table A.6 summarizes the empirical validation of each theoretical assumption underlying our scaling law. For each assumption, we specify the verification method, corresponding figure, and quantitative result.

*Table A.6.* Empirical verification of theoretical assumptions from Section 4.

| Assumption | Verification Method | Figure / Table | Result |
|---|---|---|---|
| PL (single task, Asm. 4.3) | Gradient norm vs. loss gap | Figure 3(a) | $\mu = 0.03$ |
| PL breadth (Asm. 4.3) | Aggregate $\mu$, ID / mild / strong OOD | Figure A.7 | $R^2 > 0.95$ across all violations |
| Lipschitz (Lemma 4.5) | Lindbladian distance scaling | Figure 3(b) | $C_L = 2.8$ |
| Control separation (Lemma 4.7) | Optimal control distance | Figure 3(c) | $R^2 = 0.98$ |
| $\sigma_\tau^2 \to \sigma_L^2$ | Loss variance regression | Figure A.3 | $R^2 = 0.987$ |
| $\beta \propto \eta$ (Thm. 4.8) | Learning rate ablation | Figure A.4(b) | Linear for $\eta \leq 2 \times 10^{-2}$ |

# Appendix D: Quantum Control Primer

This appendix provides background on quantum control for readers unfamiliar with the physics and specifies the Hamiltonians and parameters used in our experiments. Readers comfortable with Lindblad dynamics may skip to Section D.5.

## D.1 Quantum States and Gates

A *qubit* is a two-level quantum system with state $|\psi\rangle = \alpha|0\rangle + \beta|1\rangle$ where $|\alpha|^2 + |\beta|^2 = 1$. The state can be visualized as a point on the Bloch sphere, with $|0\rangle$ and $|1\rangle$ at the poles.

When subject to noise or partial measurement, pure states generalize to *density matrices* $\rho$, which are positive semidefinite Hermitian operators with $\text{Tr}(\rho) = 1$. A pure state $|\psi\rangle$ corresponds to $\rho = |\psi\rangle\langle\psi|$ with $\text{Tr}(\rho^2) = 1$, while mixed states have $\text{Tr}(\rho^2) < 1$.

*Quantum gates* are unitary transformations acting on qubits. The single-qubit Pauli matrices are:

$$\sigma_x = \begin{pmatrix} 0 & 1 \\ 1 & 0 \end{pmatrix}, \quad \sigma_y = \begin{pmatrix} 0 & -i \\ i & 0 \end{pmatrix}, \quad \sigma_z = \begin{pmatrix} 1 & 0 \\ 0 & -1 \end{pmatrix} \tag{A.40}$$

The $X$ gate (Pauli-$\sigma_x$) acts as a NOT operation, mapping $|0\rangle \leftrightarrow |1\rangle$.

The *quantum state fidelity* between density matrices $\rho$ and $\sigma$ is:

$$\mathcal{F}(\rho, \sigma) = \left[ \text{Tr}\sqrt{\sqrt{\rho}\,\sigma\,\sqrt{\rho}} \right]^2 \tag{A.41}$$

For a pure target state $\sigma = |\psi\rangle\langle\psi|$, this simplifies to $\mathcal{F} = \langle\psi|\rho|\psi\rangle$. Our loss function is the *infidelity* $\mathcal{L} = 1 - \mathcal{F}$.

A foundational result in quantum computing is that single-qubit rotations combined with any entangling two-qubit gate (e.g., CZ or CNOT) form a *universal gate set*: any quantum algorithm can be decomposed into these primitives (Barenco et al., 1995). This is why calibrating single- and two-qubit gates (the focus of our experiments) addresses the complete calibration problem for gate-based quantum computers.

## D.2 Open Quantum Systems and the Lindblad Equation

Real quantum processors interact with their environment, causing *decoherence*— the loss of quantum information. The Lindblad master equation governs this dissipative evolution:

$$\dot{\rho}(t) = -i[H(t), \rho] + \sum_j \mathcal{D}[L_j]\rho \tag{A.42}$$

where the *dissipator* superoperator is,

$$\mathcal{D}[L]\rho = L\rho L^\dagger - \frac{1}{2}\left( L^\dagger L\rho + \rho L^\dagger L \right). \tag{A.43}$$

Here $H(t)$ is the (possibly time-dependent) system Hamiltonian and $\{L_j\}$ are *Lindblad operators* (also called jump operators) that characterize different noise channels.

**Notation convention.** Throughout this work, we use two equivalent formulations: the main text factors rates as $\Gamma_j \mathcal{D}[\tilde{L}_j]$ with normalized jump operators $\tilde{L}_j$, while Appendix D uses $\mathcal{D}[L_j]$ with $L_j = \sqrt{\Gamma_j}\tilde{L}_j$ absorbing the rates.

For superconducting qubits, the two dominant noise channels are:

- **Relaxation** ($T_1$ decay): Energy dissipation from $|1\rangle \to |0\rangle$, characterized by the lowering operator $L_{\text{relax}} = \sqrt{\Gamma_{\text{relax}}}\,\sigma_-$ where $\sigma_- = |0\rangle\langle 1|$. The rate $\Gamma_{\text{relax}} = 1/T_1$.

- **Pure dephasing** ($T_\phi$ decay): Loss of phase coherence without energy exchange, characterized by $L_{\text{deph}} = \sqrt{\Gamma_{\text{deph}}/2}\,\sigma_z$. The factor of $1/2$ arises from the convention that $\sigma_z$ has eigenvalues $\pm 1$.

The total decoherence time satisfies $1/T_2 = 1/(2T_1) + 1/T_\phi$.

**Connection to noise power spectral density.** The dissipation rates $\Gamma_j$ arise from the qubit's coupling to environmental fluctuations. For a noise source with power spectral density $S(\omega)$, the effective rate is:

$$\Gamma = \int_0^\infty S(\omega)\,|F(\omega)|^2\,d\omega \tag{A.44}$$

where $F(\omega)$ is the *filter function* encoding the control sequence's sensitivity to noise at frequency $\omega$ (Biercuk et al., 2011). This provides the physical basis for our task parameterization: different devices have different $S(\omega)$, leading to different $\Gamma$ values.

### D.3 Single-Qubit Gate: $X$ Gate

For single-qubit experiments, we implement an $X$ gate (bit-flip, equivalent to a $\pi$ rotation about the $x$-axis of the Bloch sphere). The control Hamiltonian is:

$$H(t) = \frac{\omega_q}{2}\sigma_z + u_x(t)\sigma_x + u_y(t)\sigma_y \tag{A.45}$$

where $\omega_q$ is the qubit frequency (drift strength), and $u_x(t), u_y(t)$ are time-dependent control fields parameterized by our neural network policy $\pi_\theta$. The drift term $\frac{\omega_q}{2}\sigma_z$ represents the qubit's natural precession, while the control terms drive rotations about the $x$ and $y$ axes.

The target unitary is $U_{\text{target}} = X = \sigma_x$, which maps $|0\rangle \to |1\rangle$ and $|1\rangle \to |0\rangle$. Starting from initial state $\rho_0 = |0\rangle\langle 0|$, the target final state is $\rho_{\text{target}} = |1\rangle\langle 1|$.

**Lindblad operators.** The noise model includes relaxation and dephasing channels:

$$L_1 = \sqrt{\Gamma_{\text{relax}}}\,\sigma_-, \qquad L_2 = \sqrt{\Gamma_{\text{deph}}/2}\,\sigma_z \tag{A.46}$$

where $\sigma_- = |0\rangle\langle 1|$ is the lowering operator. The rates $\Gamma_{\text{relax}}$ and $\Gamma_{\text{deph}}$ vary across tasks according to our task distribution $p(\xi)$.

### D.4 Two-Qubit Gate: CZ Gate

For two-qubit experiments, we implement a controlled-Z (CZ) gate, which applies a $\pi$ phase to the $|11\rangle$ state while leaving other computational basis states unchanged:

$$U_{\text{CZ}} = \text{diag}(1, 1, 1, -1) = |00\rangle\langle 00| + |01\rangle\langle 01| + |10\rangle\langle 10| - |11\rangle\langle 11| \tag{A.47}$$

The CZ gate is locally equivalent to CNOT (they differ by single-qubit rotations) and is native to many superconducting architectures.

**Hamiltonian.** We model two qubits with a static ZZ coupling and individual single-qubit drives:

$$H(t) = J\,\sigma_{z,1} \otimes \sigma_{z,1} + \sum_{j=1}^{2}[u_{x,j}(t)\sigma_{x,j} + u_{y,j}(t)\sigma_{y,j} + u_{z,j}(t)\sigma_{z,j}] \tag{A.48}$$

where $J$ is the ZZ coupling strength and $\sigma_{\alpha,j}$ denotes Pauli operator $\alpha \in \{x, y, z\}$ acting on qubit $j$ (tensored with identity on the other qubit). The six control fields $\{u_{x,1}, u_{y,1}, u_{x,2}, u_{y,2}, u_{z,1}, u_{z,2}\}$ are parameterized by our neural network policy.

*Table A.7.* Single-qubit $X$ gate simulation parameters

| Parameter | Symbol | Value |
|-----------|--------|-------|
| Qubit frequency (drift) | $\omega_q$ | 1.0 (dimensionless) |
| Evolution time | $T$ | 1.0 |
| Number of control segments | $N_T$ | 20 |
| Number of control fields | — | 2 $(u_x, u_y)$ |
| Integration method | — | RK4 |
| Integration time step | $\Delta t$ | 0.005 |
| Max control amplitude | $|u|_{\max}$ | 10.0 |

**Physical intuition.** The static ZZ coupling naturally generates a relative phase between computational basis states. Under $H_0 = J\sigma_{z,1}\sigma_{z,2}$, the eigenvalues are $+J$ for $|00\rangle, |11\rangle$ and $-J$ for $|01\rangle, |10\rangle$. After time $T$, this produces phases $e^{-iJT}$ and $e^{+iJT}$ respectively. For gate time $T = \pi/(2J)$, the relative phase between $|11\rangle$ and the other states equals $\pi$, yielding the CZ gate up to single-qubit Z rotations. (Our experiments use $J = 2.0$, giving $T \approx 0.785$.)The X, Y, and Z drives provide the degrees of freedom to correct these local phases and optimize fidelity under noise.

**Noise model.** Each qubit experiences independent dephasing and relaxation:

$$\dot{\rho} = -i[H(t), \rho] + \sum_{j=1}^{2} \left[ \frac{\Gamma_{\mathrm{deph},j}}{2} \mathcal{D}[\sigma_{z,j}]\rho + \Gamma_{\mathrm{relax},j} \mathcal{D}[\sigma_{-,j}]\rho \right] \tag{A.49}$$

where $\sigma_{-,j} = |0\rangle_j\langle 1|$ is the lowering operator for qubit $j$. In our task distribution, the two qubits have correlated but not identical noise rates (within $\pm 20\%$ of each other), reflecting realistic device variation.

**Fidelity computation.** For two-qubit gates, single-state fidelity is insufficient (e.g. a control sequence might work for $|00\rangle$ but fail for superposition states). We compute average gate fidelity using 12 informationally complete input states that probe the CZ phase structure according to the following:

- X-basis products: $|++\rangle, |+-\rangle, |-+\rangle, |--\rangle$

- Y-basis products: $|+i, +i\rangle, |+i, -i\rangle$

- Mixed basis: $|1+\rangle, |1-\rangle, |+1\rangle, |-1\rangle$

- Computational basis: $|00\rangle, |11\rangle$

where $|\pm\rangle = (|0\rangle \pm |1\rangle)/\sqrt{2}$ and $|\pm i\rangle = (|0\rangle \pm i|1\rangle)/\sqrt{2}$. For each input state $|\psi_k\rangle$, we evolve $\rho_0 = |\psi_k\rangle\langle\psi_k|$ and compute

$$\mathcal{F} = \frac{1}{12} \sum_{k=1}^{12} \langle\psi_k|U_{\mathrm{CZ}}^{\dagger}\rho(T)U_{\mathrm{CZ}}|\psi_k\rangle. \tag{A.50}$$

*Table A.8.* Two-qubit CZ gate simulation parameters

| Parameter | Symbol | Value |
|---|---|---|
| ZZ coupling strength | $J$ | 2.0 (dimensionless) |
| Ideal gate time | $T$ | $\pi/(2J) \approx 0.785$ |
| Number of control segments | $N_T$ | 20–30 |
| Number of control fields | — | 6 ($u_x, u_y, u_z$ per qubit) |
| Max control amplitude | $|u|_{\max}$ | $\pi$ |
| Integration time step | $\Delta t$ | 0.01 |

## D.5 Differentiable Quantum Simulation

A key technical contribution enabling our experiments is a fully differentiable Lindblad simulator implemented in PyTorch. This allows gradients to flow from the fidelity loss through the entire quantum simulation back to the policy parameters, enabling end-to-end meta-learning.

**Integration scheme.**  We use a 4th-order Runge-Kutta (RK4) integration implementation shown below

$$k_1 = \mathcal{L}[\rho_n] \tag{A.51}$$

$$k_2 = \mathcal{L}[\rho_n + \tfrac{\Delta t}{2} k_1] \tag{A.52}$$

$$k_3 = \mathcal{L}[\rho_n + \tfrac{\Delta t}{2} k_2] \tag{A.53}$$

$$k_4 = \mathcal{L}[\rho_n + \Delta t\, k_3] \tag{A.54}$$

$$\rho_{n+1} = \rho_n + \frac{\Delta t}{6}(k_1 + 2k_2 + 2k_3 + k_4) \tag{A.55}$$

where $\mathcal{L}[\rho] = -i[H, \rho] + \sum_j \mathcal{D}[L_j]\rho$ is the Lindbladian superoperator. Each operation is implemented using standard PyTorch tensor operations, ensuring automatic differentiation compatibility.

**Piecewise-constant controls.**  The control sequence is discretized into $N_{\text{seg}}$ segments of duration $\Delta T = T/N_{\text{seg}}$. Within each segment, the control amplitudes $u_k$ are constant, and we take multiple RK4 substeps (typically $\Delta T/\Delta t \approx 5$–$10$) to maintain accuracy.

## D.6 Gradient-Based Quantum Control

The *control landscape* is the function mapping pulse parameters $\theta$ to fidelity $\mathcal{F}(\theta)$. A fundamental result in quantum control theory is that for *controllable* systems, where the Lie algebra generated by control Hamiltonians spans $\mathfrak{su}(d)$. This landscape is generically *trap-free*: all local optima are global optima (Russell et al., 2017).

**Controllability condition.**  A quantum system with static Hamiltonian $H_0$ and control Hamiltonians $\{H_1, \ldots, H_m\}$ is controllable if:

$$\text{Lie}\{iH_0, iH_1, \ldots, iH_m\} = \mathfrak{su}(d) \tag{A.56}$$

where $\text{Lie}\{\cdot\}$ denotes the Lie algebra generated by nested commutators. For our single-qubit system with $H_0 \propto \sigma_z$ and controls $\{\sigma_x, \sigma_y\}$, this is satisfied since $[\sigma_x, \sigma_y] \propto \sigma_z$.

**GRAPE.**  Gradient Ascent Pulse Engineering (Khaneja et al., 2005) exploits this favorable geometry by computing $\nabla_\theta \mathcal{F}$ via the chain rule through the quantum dynamics and applying gradient ascent. Our differentiable simulator implements the same principle using automatic differentiation, but extends it to the meta-learning setting where gradients must flow through *multiple* task adaptations.

**Open-system caveat.**  The trap-free guarantee holds rigorously for closed (unitary) systems. For open systems with Lindblad dissipation, the landscape geometry is more complex and trap-free guarantees are not generally known. However, for weak dissipation ($\Gamma T \ll 1$), the landscape remains approximately trap-free, and we verify the local Polyak-Łojasiewicz condition empirically (Figure 3(a) in the main text).

