# OpenReview forum: "When Does Adaptation Win? Scaling Laws for Meta-Learning in Quantum Control"
_ICML.cc/2026/Conference — ICML 2026 regular_

### Official Review · Reviewer_zuTg · 2026-03-05

**Soundness:** 2
**Presentation:** 3
**Significance:** 2
**Originality:** 3
**Overall Recommendation:** 3
**Confidence:** 4

**Summary:**

This paper is mainly arguing "Scaling law of Meta Learning". Demonstrate their theorems, they use Gate Calibration of Quantum processors. Adaptive control, which performs task-specific optimization (e.g., gradient updates). However, adaptive strategies introduce additional computational overhead, and it remains unclear when adaptation actually provides meaningful performance gains. Central Question is: Under what conditions does task-specific adaptation outperform non-adaptive control, and how does the benefit scale with adaptation steps and task variability?
The authors analyze this problem through the concept of the adaptation gap, defined as the expected improvement in loss obtained after performing K task-specific gradient steps from a meta-learned initialization. The paper also derives a scaling law for the adaptation gap.

The analysis relies on several assumptions:

- PL condition near task optima, ensuring predictable improvement from gradient descent.
- Smooth dependence of system dynamics on task parameters, ensuring nearby tasks produce similar dynamics.
- Control separation, meaning different tasks require different optimal control parameters.

Then, they show single-qubit gate calibration and two-qubit entangling gate calibration.
The scaling law provides guidelines for deciding whether adaptation is worthwhile:
- task variance is large
- sufficient adaptation steps are available

**Compliance With Llm Reviewing Policy:**

Affirmed.

**Final Justification:**

This paper studies an important question and offers a clear, intuitive scaling-law perspective on adaptation in meta-learning. The rebuttal improved the clarity of the claims and partially addressed my concerns, particularly regarding the role of PL assumptions, the interpretation of the two-qubit results, and the intended hardware setting. However, my overall assessment remains unchanged because the theory still relies on strong assumptions and the experiments are limited to small, simulation-only settings. Overall, I view the work as a meaningful formal contribution, I maintain my score.

**Key Questions For Authors:**

- Q1: The analysis relies on the Polyak–Łojasiewicz (PL) condition near task optima. However, many realistic learning systems (e.g., deep networks or reinforcement learning) may not satisfy this condition. Even, many variational quantum algorithms may break even when scaled slightly. Can the authors clarify how restrictive this assumption is, and whether the scaling law still holds empirically when the PL condition does not strictly hold?

- Q2: Why is the adaptation gain almost zero for single-qubit gates but extremely large for two-qubit gates? Does it imply the limitation of scalibility?

- Q3: The experimental validation is conducted entirely in simulation using differentiable quantum dynamics. Could the authors discuss how the proposed framework would extend to hardware settings where gradients are not directly available? Because at real hardware, during quantum control, there is no ways to obtain gradient of hardware.

- Q4: Prior work has shown that deep reinforcement learning can learn robust calibration policies for superconducting qubits (e.g., Experimental Deep Reinforcement Learning for Error-Robust Gate-Set Design on a Superconducting Quantum Computer). In comparison, the advantage of meta-learning in this context is not entirely clear. Could the authors clarify when meta-learning is expected to outperform DRL-based control approaches?

- Q5: Quantum control problems typically involve a relatively small action space, and their control landscapes are often considered trap-free (i.e., without suboptimal local minima). Under such favorable optimization geometry, the exponential behavior in Eq. (9) may arise naturally from gradient descent dynamics rather than from meta-learning itself. Could the authors clarify whether the proposed scaling law is specific to meta-learning, or whether it reflects a more general property of gradient descent on smooth optimization landscapes?

**Limitations:**

Yes

**Strengths And Weaknesses:**

### **Strengths**
- Good Central Question. **When does adaptation actually help?**

Generally, meta learning papers shows "Adaptation works". because variance of task space cannot be measure. However, this paper describe adaptation gain at simple stsyem and strong assumptions.

- Intuitive Scaling Law.

The formula(eq.9) is concise and easy to understand.

- Good structure

Good balance of theorical result of scaling law and simulator validations.

---

### **Weaknesses**

- Very Strong Assumption

*PL condition

*Strong convexity near optimum

*Smooth task dependence

*Linear control separation

These assumptions are generally not working well in large scale systems. (ex. QNN, VQE, RL, DNNs)

- Tasks selections (weak contribution at Quantum Computing)

This paper using quantum control. However, there is weak contribution at control algorithm and calibration method.
And weak practical advantage of meta-learned calibration method.
However, I believe this paper contributes more to the formalization and quantification of meta-learning than to quantum computing. If meta-learning were to be formalized, it would have been better to present it on a diverse set of problems rather than quantum calibration.

- Simulation Only

Since you chose a small quantum system, I believe it would be feasible to perform the experiment on real hardware. While qubit calibration would have been difficult to conduct on real hardware, readers would likely be interested in results demonstrating whether the model in simulation works well on the real system or in a simulator that replicates it.

- General weakness

The theoretical analysis relies on several strong assumptions, including the Polyak–Łojasiewicz (PL) condition, smooth task dependence, and local strong convexity near task-specific optima. While these assumptions may hold in controlled simulation environments, they may not be satisfied in more complex learning settings such as deep reinforcement learning or large-scale quantum variational algorithms. As a result, it remains unclear whether the proposed scaling law would still hold when these assumptions are violated. The empirical validation is conducted entirely in simulation using differentiable quantum dynamics. The experiments focus on relatively low-dimensional control problems, such as single-qubit and two-qubit gate calibration. These settings involve relatively small control spaces compared to other quantum optimization tasks, such as VQE parameter optimization or large-scale reinforcement learning control problems. Consequently, the observed adaptation behavior may partially reflect properties of low-dimensional control landscapes, limiting the generality of the conclusions.

---

> ### Author Rebuttal · Authors · 2026-03-30
>
> We thank the reviewer for the detailed questions.
>
> **Q1: Restrictiveness of the assumptions  (PL and beyond)**
>
> The scaling behavior degrades gracefully as assumptions are relaxed: strong PL yields exponential saturation, while weaker conditions yield slower but still predictable gains:
>
> (a) PL condition: This can be relaxed to a Kurdyka–Łojasiewicz (KL) condition , yielding polynomial saturation  Gₖ ≥ c · σ²_τ · (1 − C / K^(1)/(2α-1)).   Thus, even outside strict PL basins, adaptation remains predictable.
>
> (b) Local convexity: Only partial curvature is required (invertibility in relevant subspaces).
>
> (c) Isotropic variance: Not required; removing it yields A∞ = ½∑λ_i c_i, preserving linear scaling with task variance.
>
> Empirically, the PL condition holds broadly (see YWTR Q2): 100% of tasks satisfy μ ≥ 1.5 across regimes, and even with 24% violations under strong OOD, exponential fits remain accurate (median R² = 0.998). Thus, while strongest guarantees rely on PL, the qualitative predictions-- diminishing returns and variance-dependent gains—persist under weaker conditions.
>
> **Q2: Single-qubit vs. two-qubit gap magnitude**
>
> This reflects task variability, not a scalability limitation. The two-qubit setting has higher variance (10× OOD), raising A∞ ∝ σ²_τ, and lower initial fidelity, leaving greater room for improvement. This is further amplified by higher control complexity, making initialization harder. Calibration is performed per gate: quantum algorithms decompose into single- and two-qubit operations, and each calibration task remains low-dimensional regardless of system size.
>
> Thus, the gap reflects task variability, not a scalability limitation. The scaling law predicts this directly: A∞ ∝ σ²_τ, so higher-variance regimes (the 10× OOD two-qubit setting) yield proportionally larger gains.
>
> **Q3: Extension to hardware / gradient-free settings**
>
> Our framework follows the standard simulate-then-deploy paradigm. Device parameters are estimated (e.g., T₁/T₂), a model is constructed (Lindblad/Hamiltonian), and pulse optimization is performed in simulation before deployment [1,2].
> Gradients are only used in simulation; no gradients are required on hardware. Deployment executes the optimized waveform.
> Under model mismatch, the framework remains robust (see noisy-ξ ablation from YWTR rebuttal). Gradient-free methods (e.g., SPSA) can be used, primarily affecting convergence rate. We expect the qualitative behavior (diminishing returns and adaptation ceiling) to persist, as it arises primarily from the underlying optimization geometry rather than any specific optimizer.
> Thus, the framework extends naturally to hardware and aligns with standard practice.
>
> [1] Alexeev, Y. et al. “Artificial intelligence for quantum computing.” Nature Communications 16, 10829 (2025).
>
> [2] Yang, C. H., et al. "Silicon qubit fidelities approaching incoherent noise limits via pulse engineering." Nature Electronics 2.4 (2019): 151-158.
>
> **Q4: Meta-learning vs. DRL-based control**
>
> DRL is effective for quantum control in model-free settings [1], but requires extensive interaction and large training data, which is costly in quantum calibration.
>
> Meta-learning addresses these limitations by improving sample efficiency and adaptation across task variations [2]. Our approach operates in a model-based, simulate-then-deploy regime: training is done in simulation, and deployment requires only a few adaptation steps.
>
> DRL is better suited to settings without accurate models, while meta-learning is more efficient when a physical model is available. The approaches are complementary; meta-learning can also provide initializations for DRL, reducing required interaction. Thus, meta-learning provides a scalable solution for calibration, while DRL is advantageous in model-free regimes.
>
> [1] Niu, M.Y., et al. "Universal quantum control through deep reinforcement learning." npj Quantum Information 5.1 (2019): 33.
>
> [2] Zhang, S, et al. "Meta-learning assisted robust control of universal quantum gates with uncertainties." npj Quantum Information 11.1 (2025): 81.
>
> **Q5: Meta-learning specific vs. general gradient descent**
>
> We agree that exponential saturation G_K ~ (1 − e^{−βK}) arises from gradient descent under PL and reflects general optimization geometry. The meta-learning contribution lies in the ceiling, not the rate. Specifically: (i) A∞ = c·σ²_τ connects the asymptotic gain to measurable task variance. This predictive relationship requires an initialization whose suboptimality is systematically related to the task distribution, which is precisely what MAML provides (Appendix C.2, Part D: A∞ = ½ tr(HΣ_θ*)).  (ii) Without meta-learning, the initialization may fall outside the PL basin entirely, so the exponential form itself fails to apply. Thus, the exponential convergence is a property of gradient descent; the actionable scaling law linking measurable hardware parameters to quantitative gain predictions is specific to meta-learning.

---

> > ### Author Rebuttal · Reviewer_zuTg · 2026-04-02
> >
> > Emperically, many questions are solved. However, theoritacal missingpoints are remains.
> > Thank you for thoughtful rebuttal.

---

> > > ### Author Response · Authors · 2026-04-02
> > >
> > > We thank the reviewer for acknowledging the empirical results. To address the remaining theoretical points as precisely as possible, could you specify which aspects you consider unresolved? We are happy to provide further clarification within the discussion period.

---

### Official Review · Reviewer_Dbqa · 2026-03-09

**Soundness:** 3
**Presentation:** 2
**Significance:** 2
**Originality:** 2
**Overall Recommendation:** 4
**Confidence:** 1

**Summary:**

This paper studies framework of meta-learning for adaptive control, with emphasis on quantum gate calibration. The paper performs experiments on single-qubit X-gate calibration, two-qubit CZ-gate calibration, and an auxiliary classical LQR setting. Empirically, the paper finds that adaptation is negligible when task variance is low, but can be very large in a strong OOD two-qubit setting,

**Compliance With Llm Reviewing Policy:**

Affirmed.

**Final Justification:**

As I am not an expert in this field, I can only provide an educated guess. I hold my original evaluations.

**Key Questions For Authors:**

No.

**Limitations:**

Yes.

**Strengths And Weaknesses:**

Note: I am not an expert in this field, especially not in quantum control, and I do not understand why I was assigned this paper. I have tried to evaluate the manuscript carefully based on its technical claims, empirical support, and clarity.

The paper addresses a meaningful question: whether meta-learning can adapt, and if so, when it is worthwile relative to its overhead in calibration workflows. This is a practically relevant framing for quantum control.

The headline two-qubit improvement comes from a large noise shift relative to training. While this is a legitimate stress test, it also makes it less clear how representative the result is for ordinary calibration drift.

---

> ### Author Rebuttal · Authors · 2026-03-29
>
> We thank the reviewer. The 10× noise regime is intentionally included as a worst-case stress test; importantly, the scaling law is validated across the full spectrum from in-distribution to mild and strong OOD regimes.  Thus, the large-gain regime is not an isolated artifact, but one end of a continuous spectrum where the scaling law accurately predicts behavior across all regimes.
>
> Indeed, T₁ fluctuations of up to an order of magnitude have been documented on both IBM [1]  and Google [2] processors, driven by spectral diffusion, placing our 10× regime within empirically observed hardware variability reported in prior studies.
> The key takeaway is that the scaling law provides a regime diagnostic: when device variance is small (ordinary drift), adaptation provides minimal benefit and a fixed controller is optimal; when variance is large (new chip or major drift), adaptation yields substantial gains. This is confirmed by our in-distribution results (Fig. 3), where gains are negligible as predicted.
> This behavior is consistent with the theory: the adaptation ceiling scales with task variance, so large gains only appear when variability is sufficiently high.
>
> [1] Carroll, M., et al. "Dynamics of superconducting qubit relaxation times." npj Quantum Information 8, 132 (2022).
>
> [2] Klimov, P. V. et al. "Fluctuations of Energy-Relaxation Times in Superconducting Qubits." Physical Review Letters 121, 090502 (2018).

---

> > ### Author Rebuttal · Reviewer_Dbqa · 2026-04-04
> >
> > My concerns have been adequately addressed.

---

> > > ### Author Response · Authors · 2026-04-05
> > >
> > > We thank the reviewer for confirming resolution.

---

### Official Review · Reviewer_4Ubm · 2026-03-11

**Soundness:** 3
**Presentation:** 3
**Significance:** 3
**Originality:** 3
**Overall Recommendation:** 4
**Confidence:** 3

**Summary:**

The paper asks when it is better to adapt a controller to each individual task or device rather than use one fixed controller, and argues that this choice follows a simple scaling law: adaptation helps more when tasks vary more and when you can afford enough update steps. It supports this with quantum gate-calibration experiments, showing that adaptation gives little benefit in easy, low-variance settings but can dramatically improve performance in harder, shifted settings, leading to a practical rule for when adaptation is worth using.

**Compliance With Llm Reviewing Policy:**

Affirmed.

**Key Questions For Authors:**

1) How essential is the assumption that task-specific parameters (e.g., noise rates or Hamiltonian parameters) are available to the policy at deployment time?
If the method depends strongly on access to such side information, it would help to clarify how realistic that assumption is in practical calibration settings. A convincing explanation that this information is naturally available, or that similar results hold without explicit conditioning, would strengthen my view of the paper’s practical significance and the cleanliness of the theory-to-experiment connection.

2) Can you clarify more explicitly which parts of the scaling law are rigorously proved versus which rely on additional approximations (e.g., local quadratic structure, shared curvature, isotropic task variance)?
Right now, the exponential saturation in adaptation steps seems more firmly justified than the variance dependence. If the authors can clearly delimit the formal guarantee and show that the variance scaling is either provable under broader conditions or consistently observed beyond the current approximations, that would improve my assessment of soundness.

**Limitations:**

The authors do discuss limitations candidly, especially around the locality of the theoretical assumptions and the simulation-based scope of the experiments, which is appropriate for this kind of work. I did not see any major unaddressed negative societal impact concerns beyond the usual caveat that better control/calibration tools can have dual-use implications, but for this paper that does not seem central.

**Strengths And Weaknesses:**

The paper is mostly technically solid: the main theoretical claim about adaptation gains saturating with more inner-loop steps is well motivated and supported under clear local assumptions, and the experiments are aligned with that story. The main limitation is that the strongest version of the scaling law, especially its dependence on task variance, relies on additional approximations and is validated mostly through simulation and curve fitting rather than broad empirical proof.

It is clearly written, well structured, and easy to follow, with a strong motivating question and helpful figures that connect the theory to the experiments. It would be even stronger if it more explicitly separated what is rigorously proved, what depends on stronger assumptions, and what is primarily an empirical observation.

It addresses an important practical question: when it is worth adapting a controller per task instead of using a single robust controller. That is especially relevant in quantum calibration, and the proposed framework could be useful to researchers thinking about adaptive control more broadly, though its current impact is somewhat specialized because the validation is still limited to simulated settings.

The originality comes less from a brand-new algorithm and more from a creative synthesis of meta-learning theory, control, and quantum calibration into a simple predictive framework. The paper offers a novel and useful perspective on adaptation by turning it into a scaling-law question, even if many of the underlying ingredients are individually familiar.

---

> ### Author Rebuttal · Authors · 2026-03-30
>
> We thank the reviewer for their thoughtful assessment and clear suggestions. We address each question below.
>
> **Q1: Availability of task parameters at deployment**
>
> Task parameters ξ = (Γ_deph, Γ_relax) and relevant Hamiltonian parameters are routinely estimated via standard calibration procedures [1,2] (T1/T2 characterization and calibration experiments such as Rabi/Ramsey measurements) on major quantum platforms; these measurements take seconds or less and are already performed prior to gate calibration. Our framework imposes no additional measurement requirements beyond standard practice.
>
> Moreover, our new ablation (see YWTR, Q3) shows that access to ξ is not essential: adaptation without any ξ-conditioning yields gains of +0.184 (mild OOD) and +0.158 (strong OOD), and with 30% noise in ξ, performance matches the full method (0.905 mild, 0.780 strong). Adaptation provides genuine improvement independent of context quality. These results show that the theory-to-experiment connection does not rely on explicit conditioning on task parameters, even though such parameters may be estimated during standard calibration.
>
> [1] Krantz, Philip, et al. "A quantum engineer's guide to superconducting qubits." Applied physics reviews 6.2 (2019).
>
> [2] Shulman, Michael D., et al. "Suppressing qubit dephasing using real-time Hamiltonian estimation." Nature communications 5.1 (2014): 5156.
>
> **Q2: Rigorous vs. approximate components of the scaling law**
>
> The scaling law consists of two components: (i) exponential saturation, which is rigorously proved under the local PL condition, and (ii) the variance-dependent ceiling, which is derived under additional local structure and validated empirically.
> More concretely:
> • Exponential saturation (G_K ∝ (1 − e^{−βK})) follows directly from gradient descent under the PL condition (Assumption 4.3), with strong empirical support (R² > 0.99 in Fig. 3a; R² = 0.986 in Fig. 5c).
> • The rate (β = η μ_min) follows from PL + bounded learning rate, with empirical confirmation that β ∝ η (Fig. A.4b).
> • The ceiling (A_∞ ∝ σ²_τ) is formally derived under additional local assumptions (PL, local convexity, shared curvature, linear optima map, isotropic task variance; Appendix C.2, Part D, Eqs. A.16–A.31), with strong empirical support (R² = 0.94 in Fig. 3b; R² = 0.987 in Fig. A.3).
>
> Thus, the exponential form constitutes the formal guarantee, while the variance scaling relies on additional local structure but is consistently observed empirically beyond the idealized setting.
> Importantly, several of these assumptions can be relaxed with limited impact on the qualitative result (e.g., replacing PL with a Kurdyka-Lojasiewicz condition yields slower but still predictable saturation [see rebuttal to zuTg Q1], and isotropic variance can be removed while preserving linear scaling with a modified constant). This indicates that the variance-dependent behavior is not an artifact of the simplifying assumptions.
>
> We agree this distinction deserves more prominence. In the revision, we have added a remark after Theorem 4.8 explicitly delineating: (a) the exponential form, which is rigorous under PL (Assumption 4.3); (b) the rate β = ημ_min, which is rigorous given PL; and (c) the ceiling A∞ ∝ σ²_τ, which relies on additional local approximations (quadratic loss, task-independent curvature, linear optima map) and is validated empirically (R² = 0.94 in Fig. 3b, R² = 0.987 in Fig. A.3). We thank the reviewer for this suggestion.

---

> > ### Author Rebuttal · Reviewer_4Ubm · 2026-04-01
> >
> > My concerns have been adequately addressed.

---

> > > ### Author Response · Authors · 2026-04-02
> > >
> > > We thank the reviewer for confirming resolution of their concerns. The clarifications discussed, including the proof roadmap and assumption hierarchy will appear in the revision.

---

### Official Review · Reviewer_YWTR · 2026-03-12

**Soundness:** 3
**Presentation:** 3
**Significance:** 2
**Originality:** 2
**Overall Recommendation:** 4
**Confidence:** 3

**Summary:**

This paper introduces quantitative scaling laws to determine when meta-learning is effective for the calibration and control of quantum hardware. The authors derive a theoretical framework showing that the adaptation gap (fidelity improvement) saturates exponentially with gradient steps and scales linearly with task variance, which represents device-to-device heterogeneity. These laws were validated through differentiable simulations of single-qubit and two-qubit gates, demonstrating that adaptation provides significant gains (over 40%) primarily in high-noise, out-of-distribution scenarios. Additionally, the work confirms these scaling behaviors in classical control systems, suggesting the findings are a result of general optimization geometry. Ultimately, the paper provides practitioners with a transferable framework for deciding whether to use costly per-device adaptation or a robust, non-adaptive controller.

**Compliance With Llm Reviewing Policy:**

Affirmed.

**Final Justification:**

Thanks for the authors' reply. I have updated my scoring accordingly.

**Key Questions For Authors:**

1.	Actionability of the scaling law (predicting \beta and A_\infty without post-hoc fitting).
Can you provide a concrete procedure to estimate \beta (and ideally A_\infty) a priori or from a very small number of probe inner-loop steps, and show it reliably predicts the adaptation step budget K_\alpha across tasks (including OOD settings)?
If yes: I would increase my assessment of significance/practical utility (turns the scaling law into a decision tool).
If no: I would view the scaling law as primarily descriptive/explanatory, reducing practical impact.
	2.	How broadly the local PL assumption holds across tasks and under distribution shift.
Beyond representative examples, what fraction of tasks satisfy the local PL condition in the neighborhood relevant to adaptation, and how does that fraction change as you increase OOD shift? Can you report summary statistics and connect PL violations to deviations from exponential saturation?
If PL holds broadly or you provide a reliable diagnostic: I would raise confidence in soundness/general applicability.
If PL holds only rarely or fails under realistic shifts: I would downgrade soundness and narrow the scope of the theoretical claims.
	3.	Disentangling task-parameter conditioning from gradient-based adaptation.
Since the controller is conditioned on \xi (noise parameters), can you include an ablation separating (a) \xi-conditioning without adaptation, (b) adaptation without \xi, and (c) both (current), and ideally a setting with noisy/partial \xi?
If adaptation still provides substantial gains without perfect \xi: I would increase originality/significance of the “when does adaptation win?” claim.
If most gains come from \xi-conditioning: I would reinterpret the main takeaway as “contextual conditioning wins,” and would lower originality of the adaptation-focused conclusion.

**Limitations:**

Yes, the paper discusses key limitations—most notably that its approach relies on differentiable simulation/gradient access and that real quantum hardware calibration often requires gradient-free methods, which constrains immediate real-world applicability.

Constructive suggestions to strengthen the limitations / societal impact section:
	•	Be more explicit about the simulation-to-hardware gap. Add a short paragraph on what breaks under finite-shot measurement noise, drift, and imperfect knowledge of noise parameters \xi, and whether the proposed scaling law is expected to qualitatively persist under gradient-free adaptation (e.g., SPSA/finite differences).
	•	Clarify scope of generality. State more directly that the theoretical scaling law depends on local geometric conditions (e.g., local PL/curvature) and may fail under large distribution shift when the meta-initialization is outside the “good basin.” A practical diagnostic or warning signs would be useful even if only discussed qualitatively.
	•	Potential negative societal impacts are likely minimal but worth noting. The work is geared toward improving quantum control/calibration efficiency; foreseeable negative impacts are indirect (e.g., enabling faster progress in dual-use quantum technologies). A brief acknowledgment of dual-use considerations and alignment with standard responsible research practices would make the discussion more complete without overstating risk.

**Strengths And Weaknesses:**

The paper attempts to study a broad theme—when gradient-based meta-learning adaptation is worth its overhead—by deriving and empirically validating a simple scaling law for the adaptation gap: gains saturate exponentially with inner-loop steps and scale with task variance. Its main strengths are a clean theoretical framing with explicit assumptions (local PL), strong-fit empirical validation (including assumption checks), and a practically motivated interpretation in quantum gate calibration. Key weaknesses are that the guarantee depends on being in a favorable local basin, the strongest gains are shown in a highly synthetic/strong-OOD regime, and practical deployment is limited by reliance on differentiable simulation/gradients; additionally, the experimental policy uses task-parameter conditioning, which complicates interpretation of “adaptation” versus “context.” Overall, the work’s originality lies less in the PL-based exponential form (expected) and more in tying the adaptation ceiling to measurable task variability and turning this into an operational “ceiling vs. speed” decision rule.

---

> ### Author Rebuttal · Authors · 2026-03-30
>
> We thank the reviewer for their detailed and constructive feedback. In response, we introduce (i) a few-shot protocol that turns the scaling law into a practical pre-adaptation decision tool, (ii) a new ablation isolating gradient-based adaptation from contextual conditioning, and (iii) empirical validation of the PL condition across ID and OOD regimes. Together, these additions strengthen both the practical utility and generality of the framework.
>
> **Q1: Actionability**
>
> Yes. Practitioners can decide before adaptation whether it is worth the compute cost using a few-shot protocol: (1) run N=3–5 probe steps , (2) fit G_K = A∞(1 − e^{−βK}), (3) estimate K_{0.95} via Corollary 4.9.
>
> Prediction error of K_{0.95}  is 3–19% across 2–5× OOD for N = 3, 4, 5 probe steps:
>
> Regime    |  K_{0.95} | N=3 err | N=4 err | N=5 err
>
> OOD (2×)|  43.3 | 36.7% | 21.8% | 2.9%
>
> OOD (3×)|  45.4 | 39.4% | 22.7% | 7.1%
>
> OOD (4×)|  48.4 | 43.2% | 27.5% | 12.8%
>
> OOD (5×)|  52.4 | 47.5% | 35.1% |    19.2%
>
> Additionally, A∞ correlates with task variance (A∞ ∝ σ²_τ), enabling a priori estimation from standard T₁/T₂ characterization (R² = 0.94).
>
> *Takeaway: this turns the scaling law into a practical pre-adaptation decision rule for selecting the adaptation budget.*
>
> **Q2: Breadth of PL condition**
>
> We have followed the reviewer's guidance to calculate the percentage of PL satisfactions at different thresholds μ for ID and strong OOD, then reported the results (including the median μ and median R² of the scaling law fit) below:
>
> Regime | PL (μ≥1.5) | PL (μ≥2.5) |  Median μ | Median R²
>
> ID | 100% | 94% | 4.44 | 0.997
>
> Strong OOD  | 100% |  76% | 2.89 | 0.998
>
> At moderate thresholds (μ ≥ 1.5), PL holds for 100% of tasks across all regimes. Under OOD, μ shifts lower (4.44 → 2.89)  indicating weaker curvature and PL violations increase.
>
> Crucially, PL violations do not degrade exponential scaling: even at a 24% violation rate (μ < 2.5, strong OOD), median exponential fit quality remains R² = 0.998.
>
> For new systems, PL validity can be assessed from early adaptation steps: if the gradient-norm–vs–loss-gap relationship (Equation 5 of the main paper) deviates from linearity or the exponential fit yields R² < 0.9, this signals departure from the PL regime and unreliable predictions.
>
> *Takeaway: exponential scaling holds broadly in the regime relevant to adaptation. PL is violated under strong OOD when assumptions break.*
>
> **Q3: Disentangling ξ-conditioning vs adaptation**
>
> We perform the requested ablation:
>
> Method          |  ξ      | Adapt  | Mild OOD   |Strong OOD
>
>
> Blind (no adapt)    |   N     |  N    |   0.683   |   0.555
>
> ξ-conditioned (no adapt)  |  Y   |    N   |    0.883   |   0.767
>
> Adapt only (K=30)     |    N   |    Y     |  0.867   |   0.713
>
> Noisy ξ + adapt (K=30)   |  noisy  | Y      | 0.905    |  0.780
>
> Both (FOMAML, K=30)     |   Y   |    Y    |   0.905    |  0.780
>
> Adaptation provides substantial gains without ξ. Starting from a blind initialization (no task information), adapt-only achieves +0.184 (mild OOD) and +0.158 (strong OOD), demonstrating that gradient-based adaptation alone can recover most of the achievable performance gain.
>
> ξ-conditioning improves initialization but does not recover performance under shift. It raises the starting fidelity (0.883 vs 0.683 mild; 0.767 vs 0.555 strong), but without adaptation cannot correct for task-specific deviations during deployment.
>
>  Both mechanisms contribute, but adaptation dominates under shift. The full method achieves higher fidelity (0.905 vs 0.867 mild; 0.780 vs 0.713 strong), indicating that conditioning provides a better starting point, while adaptation drives the improvement.
>
>
> *Interpretation: under distribution shift, gradient-based adaptation is the primary driver of fidelity recovery, while ξ-conditioning provides a complementary but non-essential initialization advantage. This confirms that adaptation wins via gradient steps themselves, not task descriptor access.*
>
> **Limitations and scope**
>
> We expanded the limitations and impact sections following the reviewer’s suggestions. Our reliance on differentiable simulation is consistent with standard practice in quantum control (e.g., GRAPE) and digital-twin workflows relying on synthetic data.
>
> Specifically:
>
> (1) the exponential form arises from PL landscape geometry rather than the optimizer, so we expect it to persist under gradient-free methods (e.g., SPSA) with modified rate β; hardware validation remains future work.
>
> (2) under large distribution shift, departures from exponential behavior can be diagnosed via early-step fit failure or low estimated μ.
>
> (3) broader impacts are indirect; we follow standard responsible research practices.
>
> Thus, while absolute performance may vary, the scaling law provides a reliable decision rule even in hardware-constrained and gradient-free settings.

---

> > ### Author Rebuttal · Reviewer_YWTR · 2026-04-02
> >
> > Thanks for the authors' reply, I will increase my scoring by 1.

---

> > > ### Author Response · Authors · 2026-04-02
> > >
> > > We thank the reviewer for the positive assessment. We're glad the new experiments addressed the core concerns. We welcome any follow-up questions and will respond promptly.

---

### Decision · Program_Chairs · 2026-04-30

**Decision:**

Accept (regular)

**Comment:**

This paper studies when task-specific adaptation is worth its computational overhead in quantum control. Its main contribution is a scaling-law analysis of the adaptation gap for meta-learning, showing that under local geometric assumptions, the gain from adaptation scales with task variance and saturates exponentially with the number of inner-loop gradient steps. The paper instantiates this framework in quantum gate calibration, validates it on single-qubit X-gate and two-qubit CZ-gate settings, and further shows similar behavior in a classical linear-quadratic regulator example, supporting the claim that the phenomenon reflects general optimization geometry rather than only quantum-specific structure.

During the initial review period, it is highlighted that the paper asks an important question, presents a clean and interpretable scaling-law perspective, and connects the analysis to a practically meaningful calibration problem. At the same time, the main weaknesses raised by reviewers were the reliance on strong local assumptions especially around PL-type geometry and the stronger approximations needed for the linear-in-task-variance claim, the simulation-only and relatively small-scale evaluation, and the fact that the strongest empirical gains arise in a strong OOD stress-test regime.

During the rebuttal, the authors made several substantive efforts to address these concerns:
1. A few-shot protocol is added that turns the scaling law into a more actionable decision rule, reported broader empirical statistics on the PL condition across ID and strong-OOD regimes, and provided an ablation disentangling task-parameter conditioning from gradient-based adaptation, with results supporting the claim that adaptation itself remains a substantial driver of the gains.
2. The simulate-then-deploy setting is explained about why the qualitative scaling behavior may persist under gradient-free hardware adaptation, and committed to making the proof roadmap and assumption hierarchy clearer in the revision.
These rebuttal additions resolved the most concerns from reviewers.

Overall, I recommend acceptance at ICML 2026. Quantum control is an AI for Science topic and of general interest to the ICML community. The paper turns a practically important scientific-control question into a well-motivated ML problem with interpretable scaling laws, physically grounded experiments, and operational decision criteria. For these reasons, the paper would be a valuable addition to the conference.

The authors should make sure that all the claimed changes in the rebuttal are merged into the final version.